# Strategies for Pretraining Neural Operators

**Anthony Zhou**[1]
*Department of Mechanical Engineering*
*Carnegie Mellon University*

*ayz2@andrew.cmu.edu*

**Cooper Lorsung**[1]
*Department of Mechanical Engineering*
*Carnegie Mellon University*

*clorsung@andrew.cmu.edu*

**AmirPouya Hemmasian**
*Department of Mechanical Engineering*
*Carnegie Mellon University*

*ahemmasi@andrew.cmu.edu*

**Amir Barati Farimani**[*]
*Department of Mechanical Engineering*
*Department of Machine Learning*
*Carnegie Mellon University*

*barati@cmu.edu*

**Reviewed on OpenReview:** *https://openreview.net/forum?id=9vEVeX9oIv*

## Abstract

Pretraining for partial differential equation (PDE) modeling has recently shown promise in scaling neural operators across datasets to improve generalizability and performance. Despite these advances, our understanding of how pretraining affects neural operators is still limited; studies generally propose tailored architectures and datasets that make it challenging to compare or examine different pretraining frameworks. To address this, we compare various pretraining methods without optimizing architecture choices to characterize pretraining dynamics on different models and datasets as well as to understand its scaling and generalization behavior. We find that pretraining is highly dependent on model and dataset choices, but in general transfer learning or physics-based pretraining strategies work best. In addition, pretraining performance can be further improved by using data augmentations. Lastly, pretraining can be additionally beneficial when fine-tuning in scarce data regimes or when generalizing to downstream data similar to the pretraining distribution. Through providing insights into pretraining neural operators for physics prediction, we hope to motivate future work in developing and evaluating pretraining methods for PDEs.

## 1 Introduction

Pretraining is an immensely popular technique in deep learning in which models learn meaningful context from a large dataset and apply this knowledge to downstream tasks (Devlin et al., 2019; Chen et al., 2023; Schiappa et al., 2022). In particular, recent work has highlighted the importance of self-supervised learning, which can leverage the inherent structure of unlabeled data and learn meaningful latent representations (Bardes et al., 2022; Chen et al., 2020; Leyva-Vallina et al., 2023; He et al., 2021). The success of these self-supervised pretraining strategies has motivated their application to broad scientific and engineering problems (Wang et al., 2022a;b; Cao et al., 2023; Zhou & Farimani, 2024a; Meidani et al., 2024; Li et al., 2024; Nguyen et al., 2023). In particular, pretraining has been used in partial differential equation (PDE) modeling to improve neural operators and evaluate their scalability and generalizability (McCabe et al., 2023; Hao et al., 2024).

---

[1]Equal Contribution. [*]Corresponding Author.

Neural operators for PDEs have gained substantial interest in recent years due to their ability to quickly predict physics through inference (Li et al., 2021; Lu et al., 2021a; Brandstetter et al., 2023). Despite potential speed gains, neural operators currently struggle to generalize to unseen physics, and initial training can be slow (Lu et al., 2022; Gupta & Brandstetter, 2022). To address this issue, many works have explored different strategies to improve generalization by incorporating additional system information (Lorsung et al., 2024; Liu et al., 2023b; Takamoto et al., 2023) and pretraining neural operators across large, diverse physics to quickly fine-tune to solve PDEs (Hao et al., 2024; McCabe et al., 2023; Shen et al., 2024; Hang et al., 2024; Goswami et al., 2022). Despite showing good performance, these works usually require the use of tailored neural operators and datasets to learn different physics. This contrasts with broader deep learning trends in which pretraining methods can universally benefit models; for example, pretraining losses that are applied across CNN models (Noroozi & Favaro, 2016; Lee et al., 2017) or GNN models (Hu et al., 2020). As a result, in this work, we consider existing pretraining frameworks, as well as propose novel methods for pretraining PDE models that are flexible and can be applied across architectures or datasets.

By considering pretraining methods that are model agnostic, we can provide a detailed and level comparison of pretraining methods on a shared experimental setup. To our knowledge, this is the first work that makes an effort to compare pretraining strategies without tailored architecture choices, which allows an understanding of how pretraining affects learning in different regimes. Specifically, we compare different pretraining strategies and consider the effect of PDE data augmentations, a popular technique to improve pretrained model performance (He et al., 2021; Xie et al., 2022; Zhou & Farimani, 2024b; Brandstetter et al., 2022). Additionally, we study the performance of pretrained models with scarce fine-tuning data as well as their generalization behavior to unseen coefficients or PDEs.

Through this work, we hope to broaden the understanding of how neural operators can be pretrained for physics prediction. We organize existing pretraining strategies, propose novel vision-inspired strategies, and include common pretraining baselines to assemble a broad set of methods for learning PDE representations. We focus on vision-inspired strategies because the point-wise representation of system values on a grid for a PDE is analogous to pixel-based representations of images, and the temporal structure of PDE system evolution is analogous to videos. We find that PDE pretraining varies depending on model and dataset choice, but in general using transfer or physics-based pretraining strategies work well. In addition, transformer or CNN-based architectures tend to benefit more from pretraining than vanilla neural operators. Furthermore, the use of data augmentations consistently improves pretraining performance in different models, datasets, and pretraining strategies. Lastly, we find that pretraining is more beneficial when fine-tuning in low-data regimes, or when downstream data is more similar to pretraining data. We hope that these insights can be used to guide future work in the development and evaluation of pretraining methods for PDEs. We make code available here: `https://github.com/anthonyzhou-1/pretraining_pdes` , and use 2D PDE datasets from Zhou & Farimani (2024b) which can be found here: `https://zenodo.org/records/13355846` .

## 2 Related Works

The field of neural operators has grown rapidly in recent years, with many architectures developed to accurately approximate solutions to PDEs (Li et al., 2021; 2023a; Gupta & Brandstetter, 2022; Brandstetter et al., 2023; Lu et al., 2021a). Many works expanded on this to propose architectures to solve PDEs more quickly, with less compute, or on irregular grids (Li et al., 2023b; Hemmasian & Barati Farimani, 2023; Li et al., 2023c), and as a result, within a range of test problems, neural operators can solve PDEs quickly and accurately. However, neural operators still struggle to generalize across diverse physics, and as a result many approaches have been developed to pretrain neural operators. We summarize these past works in Table 1, and briefly describe the main approaches here. Furthermore, we review additional work from the broader field of pretraining and transfer learning due to its relevance to PDE problems.

### 2.1 PDE Transfer Learning

Many past works consider transferring knowledge between PDE parameters and domains as a form of pre-training. These works often design specific architectures that are tailored for transferring weights or layers between tasks. For example, Goswami et al. (2022) design task-specific layers of a DeepONet to be used

| Category | PDEs | Characteristic | Reference |
|---|---|---|---|
| Transfer | Darcy, Elasticity
Poisson, INS
Poisson, INS, Wave, FP
Heat, Adv, Nag, Burg, NS, AC
NS, Elasticity | Fine-tuning task layers to transfer between domains
Direct transfer across PDE domains
Desiging a transferrable model by modifying neurons
Combining operator modules with gating
Pos. encoding and masked pretraining across PDEs | Goswami et al. (2022)
Chakraborty et al. (2022)
Zhang et al. (2023c)
Tripura & Chakraborty (2023)
Rahman et al. (2024) |
| Large Models | Poisson, Helm
CNS, SWE, DiffReact
INS, CNS, SWE, DiffReact
Adv, Burg, Diff-Sorp, SWE, NS
INS, CNS, SWE, DiffReact
Poisson, Helm, NS, Wave, AC | Scaling model and datasets to characterize transfer
Embedding PDEs to a common space, with Axial ViT
Denoising & Fourier attention with large models/data
Aligning LLM guidance across diverse PDEs
Conditional transformer across large PDE datasets
Scaling operator transformers to large datasets | Subramanian et al. (2023)
McCabe et al. (2023)
Hao et al. (2024)
Shen et al. (2024)
Hang et al. (2024)
Herde et al. (2024) |
| Contrastive | KdV, Burg, KS, INS
Heat, Advection, Burg
Burg, Adv-Diff, NS | Lie Symmetries for contrastive learning
Physics-informed distance in a contrastive framework
Physical invariances to contrastively learn an encoder | Mialon et al. (2023)
Lorsung & Farimani (2024)
Zhang et al. (2023b) |
| Meta-Learning | HGO, Elasticity, Tissue
LV, GS, NS
LV, GS, GO, NS | Model-agnostic meta-learning loss across tasks
Novel loss term to maximize learning between PDEs
Hyper-network to adapt operators for specific tasks | Zhang et al. (2023a)
Yin et al. (2021)
Kirchmeyer et al. (2022) |
| In-Context | Poisson, Helm, DiffReact, NS
Poisson, DiffReact | Masked pretraining and in-context learning for PDEs
In-context learning by prompting a transformer | Chen et al. (2024)
Yang et al. (2023) |

Table 1: A review of past works on pretraining neural operators for PDEs. We organize works by approximate categories and describe their data and methods.

with different domains of 2D Darcy Flow and Elasticity problems. Another approach proposed by Tripura & Chakraborty (2023) is to design different operators that learn specific PDE dynamics and combine these in a mixture of experts approach, motivated by the observation that PDEs can often be compositions of each other. To address the issue of transferring between physical domains that can have different numbers of variables, Rahman et al. (2024) extend positional encodings and self-attention to different codomains/channels.

## 2.2 Large PDE Modeling

An extension of transfer learning is to train large models on diverse physics datasets, with the intention of learning transferable representations through scaling behavior (Wei et al., 2022; Kaplan et al., 2020; Brown et al., 2020). 61 initially explores this scaling behavior by training large neural operator models on large PDE datasets to evaluate its ability to adapt to different coefficients. McCabe et al. (2023) propose a tailored architecture for solving problems across different physics, and Hao et al. (2024) expand on this by making architectural advancements and training on more diverse physics. Despite different approaches and datasets, these works generally rely on tailored, scalable architectures for large PDE datasets; pretraining is framed as physics prediction across diverse physics and fine-tuning is done on the pretraining distribution or on unseen coefficients/PDEs.

## 2.3 PDE Contrastive Learning

Following the success of contrastive learning in the vision domain (Chen et al., 2020; Bardes et al., 2022; Zbontar et al., 2021), various methods for PDE contrastive learning have been proposed. Mialon et al. (2023) propose a contrastive learning framework in which augmented PDE samples are represented in a similar way in latent space; notably augmentations are done with physics-preserving Lie augmentations (Brandstetter et al., 2022). Zhang et al. (2023b) follow a similar approach in which physically invariant samples are clustered together in latent space, while Lorsung & Farimani (2024) rely on PDE coefficients to define a contrastive loss. In general, contrastive methods have extensive literature and theory, however they tend to be challenging to pretrain and may have incremental gains in the PDE domain.

## 2.4 Meta/In-context Learning for PDEs

Additional past work considers adapting meta-learning (Finn et al., 2017) paradigms from the broader deep learning community to the PDE domain. Zhang et al. (2023a) consider a direct adaptation of model-agnostic meta-learning to PDE tasks, while Yin et al. (2021) and Kirchmeyer et al. (2022) apply novel losses and

architectures to maximize shared learning across different tasks. Following in-context learning trends of transformer models (Dong et al., 2023), Chen et al. (2024) and Yang et al. (2023) explore using in-context learning to prompt models with PDE solutions to generalize to unseen PDE coefficients.

### 2.5 Pretraining and Transfer Learning

While PDE problems contain unique structure and challenges, works from the broader field of pretraining and transfer learning can still be leveraged to inform hypotheses and inspire novel work. In particular, many previous works seek to understand how models can learn representations that transfer across tasks or domains, as well as design architectures and pretraining tasks to optimize this (Jiang et al., 2022; Liu et al., 2023a; Zhuang et al., 2020). Under this framework, most previous work in the PDE domain is considered supervised pretraining, with the notable exception of contrastive methods. Additionally, novel works tend to focus on improving PDE-specific architectures, rather than designing specific objectives for task or domain transfer. We hypothesize that this is a reflection of the need for numerical accuracy in physics problems; prior work generally focuses on empirically improving prediction accuracy on different benchmark PDEs rather than leveraging theoretical justifications. This could highlight potential future opportunities to examine more complex or principled pretraining learning methods such as adversarial learning or causal learning.

## 3 Methods

### 3.1 Data Augmentations

Following the prevalence of data augmentation in the broader deep learning community (Chen et al., 2020; Perez & Wang, 2017), we consider the use of data augmentations adapted to the PDE domain.

#### 3.1.1 Lie Point Symmetry Data Augmentations

We consider a recent body of work proposing Lie Point Symmetry Data Augmentations (Brandstetter et al., 2022; Mialon et al., 2023), a set of PDE-specific data augmentations that preserve the underlying dynamics. Mathematically, given a PDE, one can derive a set of transformations $\{g_1, g_2, ..., g_n\}$, each with a parameter $\{\epsilon_1, \epsilon_2, ..., \epsilon_n\}$ that can be randomly sampled to modulate the strength of the transformation. Since some PDEs may exhibit more Lie symmetries than others, we consider only shifting the PDE solution in space (*Shift*), which is valid for all PDEs considered, to ensure a fair comparison between datasets. For further details on mathematical theory and its implementation in augmenting PDEs, we refer the reader to Mialon et al. (2023) and Olver (1986).

#### 3.1.2 Physics-Agnostic Data Augmentations

In computer vision literature, many successful data augmentations heavily modify inputs (Chen et al., 2020); in particular, cropping and cutting out portions of an image would not respect physics if adapted to the PDE domain. Following this, we investigate the effect of data augmentations that are physics-agnostic, in that they can be applied to any PDE since the augmentation does not preserve the underlying dynamics. Following recent work on denoising neural operator architectures (Hao et al., 2024), we consider adding Gaussian noise during pretraining (*Noise*). Furthermore, we consider scaling the PDE solution (*Scale*), in which the PDE solution values are scaled by a random constant, an approach similar to a color distortion, whereby the hue, contrast, saturation, or lightness of an image is scaled. Although these visual characteristics do not have direct counterparts in the PDE domain, their scaling can interpreted as scaling the intensity of a PDE solution. For certain simple PDEs, scaling can preserve physics, but this is not generally true due to nonlinearities in more complex PDEs, such as the Burgers equation or Navier-Stokes equations. Additional details on hyperparameters and the implementation of data augmentations can be found in Appendix D.4.

### 3.2 Pretraining Strategies

In this work, we consider using pretraining strategies that are agnostic to the neural operator architecture to ensure compatibility with different applications and future architecture advances, and describe them in

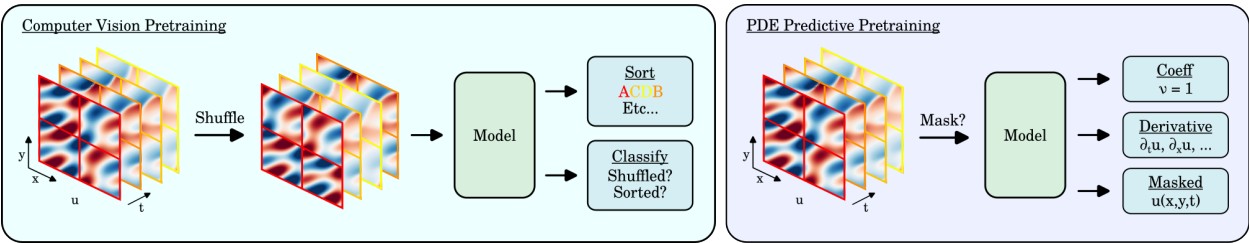

Figure 1: An illustration of pretraining strategies adapted from computer vision (CV) and predicting PDE characteristics. **Left:** CV methods can be described by different shuffling mechanisms and losses. *Binary* pretraining only classifies if a sequence is shuffled or not, while *TimeSort, SpaceSort* and *Jigsaw* sort sequences shuffled in various ways, either along the spatial, temporal, or combined dimensions. **Right:** PDE data has inherent structure that can be leveraged to predict underlying characteristics. Coefficients of the PDE can be regressed, as well as its spatial and temporal derivatives. Additionally, inputs can be masked to regress the solution field $u$ and learn underlying dynamics.

Figure 1. This approach is also consistent with the broader computer vision domain, where models are fully shared between pretraining and downstream tasks and can be adapted to different architectures (e.g. CNN, ViT) (Chen et al., 2020; Xie et al., 2022; He et al., 2021). We provide further details on design considerations and the implementation of pretraining strategies in Appendix D.3.

### 3.2.1 Computer Vision Strategies

Inspired by diverse pretraining strategies to learn image representations, we adapt many pretraining strategies from the computer vision (CV) domain to the PDE domain. In general, these strategies aim to train models through predicting visual attributes or sorting spatio-temporal sequences to learn visual representations without labels.

Firstly, we consider an early work that pretrains a model to verify if a video is in the correct temporal order (Misra et al., 2016). This problem is formulated as a binary classification task in which a shuffled video and the original video are assigned separate labels; within this work, we refer to this as *Binary* pretraining. Subsequent work proposed methods that not only verify temporal order, but can also sort temporally shuffled video frames (Lee et al., 2017). This is generally formulated as a $n-$way classification task, where $n$ denotes the number of permutations in which a sequence of frames can be sorted. In the context of physics data, we can opt to shuffle the data spatially or temporally, as such we refer to these two pretraining strategies as *TimeSort* or *SpaceSort*. Empirically, *SpaceSort* does not perform well, so we omit this strategy from our results and analysis. For completeness, we include a preliminary set of results demonstrating this in Appendix E.

An extension of sorting samples that have been shuffled along a single dimension (e.g., time, space) is to sort samples shuffled across all dimensions. For images, sorting images shuffled along both the $x$ and $y$ axes is implemented by solving jigsaw puzzles, a challenging task that reassembles an image from its shuffled patches (Noroozi & Favaro, 2016). This work has been extended to the video domain by solving spatio-temporal puzzles (Kim et al., 2018). The extension to PDE data requires sorting data that have been partitioned into discrete patches and shuffled along the space and time axes; we refer to this strategy as *Jigsaw*. One issue is that the number of possible classes scales with the factorial of the number of patches, and many shuffled sequences are not significantly different from each other. To mitigate this, we sample the top $k$ shuffled permutations that maximize the Hamming distance between the shuffled and the original sequence (Noroozi & Favaro, 2016); this ensures that models can see diverse samples during pretraining while limiting the number of classes in the pretraining task.

### 3.2.2 PDE Predictive Strategies

Within the PDE domain, there are physics-specific characteristics that PDE data exhibit that can be leveraged for pretraining; this is analogous to predicting motion or appearance statistics in vision pretraining

tasks (Wang et al., 2019; Yao et al., 2020). One strategy considers the fact that PDE data depends on equation variables and coefficients, and predicting these coefficients from the PDE data could be useful. This is implemented as a regression task, where the coefficient values are regressed from a snapshot of PDE data; we refer to this strategy as *Coefficient*.

Additionally, PDE data can be described by the derivatives of current physical values. For example, many finite difference schemes rely on spatial and temporal derivatives of the current vector or scalar field to advance the solution in time. Inspired by this, we propose a pretraining strategy that predicts the spatial and temporal derivatives of PDE data. For 2D PDEs, this is implemented as a regression tasks where the fields $(u_x, u_y, u_{xx}, u_{yy}, u_t)$ are regressed from a solution $u$; we refer to this strategy as *Derivative*.

Lastly, numerical solutions of PDEs tend to leverage information of local relationships to solve equations. For example, finite difference schemes use information from neighboring nodes to calculate spatial derivatives. Motivated by this, we propose a pretraining strategy that randomly masks data in space and time and uses this incomplete information to reconstruct the full solution. This is implemented by patching the solution in space and time, randomly replacing masked patches with a learnable mask token, and regressing the true solution; we refer to this strategy as *Masked*. We adapt mask tokens to be learnable in physics space, rather than embedding space, in order to be used by any model beyond just transformers.

### 3.2.3 Contrastive Strategies

A common strategy for pretraining in computer vision domains is to exploit similarities in the data to align samples in latent space. A proposed strategy to do this for PDEs is Physics Informed Contrastive Learning (PICL), which uses a Generalized Contrastive Loss (Leyva-Vallina et al., 2023) to cluster PDE data based on their coefficients in latent space (Lorsung & Farimani, 2024). Another strategy for self-supervised learning of PDE dynamics is using an encoder to align Lie augmented or physically invariant latent PDE samples (Mialon et al., 2023; Zhang et al., 2023b). Both works require the use of a specific encoder along with the neural operator backbone, since a separate model is contrastively pretrained to provide conditioning information to a backbone for physics prediction. To adapt these strategies to our experimental setup, we consider directly pretraining the neural operator contrastively with these strategies. However, these methods did not seem to show significant improvements over no pretraining, as such, they are omitted from our main results and analysis. Interested readers are directed to Appendix E for preliminary results regarding these methods.

## 4 Experiments

To evaluate the effectiveness of the proposed pretraining strategies and data augmentations, we consider a diverse set of experiments and neural operator architectures to train on. In particular, we hope to understand whether different architectures or datasets influence pretraining performance and construct a holistic view of pretraining for diverse PDE applications. We provide an overview of the setup and the different experiments possible in Figure 2.

### 4.1 Data

We consider predicting physics for the 2D Heat, Advection, Burgers, and incompressible Navier-Stokes equations. These equations describe a diverse range of fluid phenomena and form tasks of varying difficulties. For our experiments, we consider pretraining on a combined set of 2D Heat, Advection, and Burgers data, which contain 9216 data samples (3072 for each equation), as well as fine-tuning on a smaller set of 1024 unseen samples for each PDE. We only pretrain on the Heat, Advection, and Burgers equations since the numerical data for these PDEs are easier to generate, and as a result, transferring pretrained knowledge to more challenging PDEs can be evaluated as a potentially useful method.

### 4.1.1 Heat, Advection, and Burgers Equations

The 2D Heat, Advection, and Burgers equations are given by:

$$\partial_t u - \nu \nabla^2 u = 0, \quad \text{Heat} \tag{1}$$

$$\partial_t u + \mathbf{c} \cdot \nabla u = 0, \quad \text{Advection} \tag{2}$$

$$\partial_t u + u(\mathbf{c} \cdot \nabla u) - \nu \nabla^2 u = 0, \quad \text{Burgers} \tag{3}$$

To ensure a diverse set of physics data, the equation coefficients are randomly sampled according to Zhou & Farimani (2024b). In particular, for the Heat equation, we sample $\nu \in [2 \times 10^{-3}, 2 \times 10^{-2}]$, for the Advection equation, we sample $\mathbf{c} = [c_x, c_y] \in [0.1, 2.5]^2$, and for the Burgers equation, we sample $\nu \in [7.5 \times 10^{-3}, 1.5 \times 10^{-2}]$, and $\mathbf{c} = [c_x, c_y] \in [0.5, 1.0]^2$; we refer to this dataset as in-distribution (*In*). Since these equations also comprise the pretraining set, we additionally consider a case where the downstream dataset comes from a separate distribution; in this case, we sample $\nu \in [2 \times 10^{-2}, 3 \times 10^{-2}]$ for the Heat equation, $\mathbf{c} = [c_x, c_y] \in [2.5, 3.0]^2$ for the Advection equation, and $\nu \in [5.0 \times 10^{-3}, 7.5 \times 10^{-3}]$, and $\mathbf{c} = [c_x, c_y] \in [1.0, 1.25]^2$ for the Burgers equation. We refer to this dataset as out-of-distribution (*Out*).

In all cases, periodic boundary conditions are enforced and the solution is solved in a domain $(x, y) = [-1, 1]^2$ from $t = 0$ to $t = 2$. Furthermore, initial conditions are randomly from a summation of sine functions; the parameters are uniformly sampled from from $A_j \in [-0.5, 0.5], \omega_j \in [-0.4, 0.4], l_{xj} \in \{1, 2, 3\}, l_{yj} \in \{1, 2, 3\}, \phi_j \in [0, 2\pi)$ while fixing $J = 5, L = 2$:

$$u(0, x, y) = \sum_{j=1}^{J} A_j sin(2\pi l_{xj} x/L + 2\pi l_{yj} y/L + \phi_j) \tag{4}$$

For additional information on data splits and numerical methods, we refer readers to Appendix D.1.

### 4.1.2 Incompressible Navier Stokes Equations

The incompressible Navier Stokes equations are considered for fine-tuning pretrained models to predict more challenging physics. To ensure consistency between the pretraining and fine-tuning tasks, we use the vorticity form of the Navier-Stokes equation in order to predict a scalar field following the setup in Li et al. (2021):

$$\partial_t \omega + \mathbf{u} \cdot \nabla \omega - \nu \nabla^2 \omega = f(x, y), \quad \nabla \cdot \mathbf{u} = 0, \quad \nabla \times \mathbf{u} = \omega \tag{5}$$

$$f(x, y) = A(sin(2\pi(x + y)) + cos(2\pi(x + y))) \tag{6}$$

We formulate this problem with periodic boundary conditions, variable viscosity $\nu$, and variable forcing function amplitude $A$. Specifically, the viscosity is sampled uniformly from $\nu \in \{\{1, 2, 3, 4, 5, 6, 7, 8, 9\} \times 10^{-\{6,7,8,9\}}\}$ and the amplitude is uniformly sampled from $A \in \{\{1, 2, 3, 4, 5, 6, 7, 8, 9, 10\} \times 10^{-3}\}$. The data is generated in a domain $(x, y) = [0, 1]^2$ and from $t = 0$ to $t = 7.75$, following the setup from Lorsung et al. (2024); furthermore, the initial conditions $\omega_0$ are generated from a Gaussian random field according to Li et al. (2021).

### 4.2 Neural Operators

To compare different pretraining and data augmentation strategies, we consider their effects on improving the PDE prediction performance of different neural operators. Specifically, we consider the neural operators: Fourier Neural Operator (FNO) (Li et al., 2021), DeepONet (Lu et al., 2021a) and OFormer (Li et al., 2023a). Additionally, we consider the Unet model; while it is not explicitly a neural operator, it is commonly used in literature and has shown good performance (Ronneberger et al., 2015; Gupta & Brandstetter, 2022). These neural operators are first trained using a pretraining strategy before being fine-tuned on a PDE prediction task; this could either be fixed-future prediction to model a static solution or autoregressive prediction to

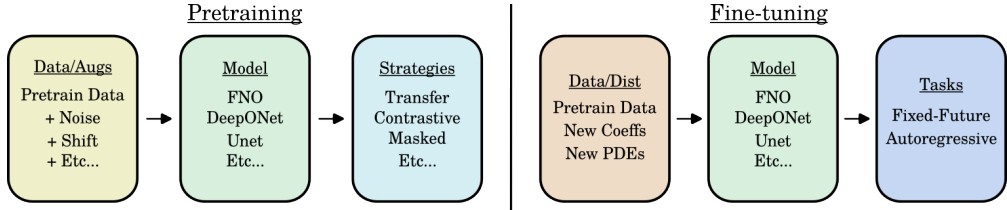

Figure 2: **Experimental Setup**. During pretraining we consider different data augmentations, model choices, and pretraining tasks and evaluate their downstream performance through fine-tuning on physics prediction tasks. During fine-tuning, we leverage the same pretrained model to improve fixed-future or autoregressive prediction on the same pretraining data distribution, unseen coefficients, or new PDEs. Through this setup we can explore a wide variety of pretraining strategies and augmentations and quantify their effects on different models, PDEs, datasets, and tasks.

model a time-dependent solution. In all experiments, prediction tasks are formulated using only solution field values and grid information. Additional details on the model hyperparameters and implementation can be found in Appendix D.2.

### 4.3 Pretraining Strategies

We compare models pretrained with different strategies with a baseline model that has not been pretrained (*None*) as well as a model trained with the same physics prediction objective on the pretraining dataset, more commonly known as transfer learning (*Transfer*). Furthermore, we vary the size of the fine-tuning dataset to study the effects of pretraining when given scarce downstream data. The fine-tuning dataset is also varied between data samples that are within the pretraining distribution (*In*), outside the pretraining distribution with respect to the PDE coefficients (*Out*), or on samples from an unseen PDE (*NS*). Lastly, we study the effects of adding data augmentations during pretraining and fine-tuning.

### 4.4 Data Augmentation

Data augmentation is implemented by doubling the pretraining and fine-tuning data, where each sample has a 50% chance of being augmented. Our noise augmentation adds a small amount of Gaussian noise to each frame independently, while our shift and scale augmentations are applied uniformly to the entire trajectory.

### 4.5 Fixed Future and Auto-regressive Prediction

To model physics problems with static solutions, we consider predicting a PDE solution field at a fixed timestep after an initial snapshot of the PDE data. In particular, given the PDE data from $t = 1$ to $t = 8$, models are trained to predict the PDE solution at $t = 32$.

Alternatively, to model physics problems with time-dependent solutions, we consider auto-regressively predicting PDE solutions directly after a current snapshot of PDE data. This is implemented using PDE data on the interval $[t, t + 8)$ as an input to predict future PDE solutions on the interval $[t + 8, t + 16)$. In addition, we use the pushforward trick (Brandstetter et al., 2023) to stabilize training. This introduces model noise during training by first predicting a future time window from ground-truth data and then using this noisy prediction as a model input; importantly, no gradients are propagated through the first forward pass. Additional details on training parameters can be found in D.5.

## 5 Results

We now systematically benchmark our pretraining and data augmentation strategies, as well as their combination. Presented below are results on our autoregressive task. Fixed-future results are given in appendices

Table 2: **Effects of Pretraining for Auto-regressive Prediction:** We present comparisons of different pretraining strategies after pretraining on the Heat, Advection, and Burgers equations and fine-tuning in 500 unseen samples.The insights are distilled into two tables, for full results see Appendix A.

(a) **The best pretraining strategy varies with model and dataset choice.** We compare the highest performing pretraining strategies on autoregressive prediction; although performance varies widely, transfer learning performs well in many settings.

| Model | Best Pretraining Method | | | |
| | Heat | Advection | Burgers | NS |
| --- | --- | --- | --- | --- |
| FNO | Derivative | Transfer | PICL | None |
| DeepONet | PICL | Transfer | Transfer | Transfer |
| OFormer | Transfer | PICL | Transfer | None |
| Unet | Transfer | TimeSort | Transfer | Transfer |

(b) **Different models display different benefits from pretraining.** We compare the improvement of the highest performing pretraining strategy to no pretraining. The models show different capacities to be pretrained.

| Model | Improvement w/ Best Strategy | | | |
| | Heat | Advection | Burgers | NS |
| --- | --- | --- | --- | --- |
| FNO | 14.43% | 7.459% | 1.430% | 0.000% |
| DeepONet | 3.580% | 1.852% | 15.74% | 2.894% |
| OFormer | 38.91% | 4.594% | 17.12% | 0.000% |
| Unet | 29.16% | 1.899% | 9.706% | 1.862% |

A and B and generally show the same trends as our autoregressive results. We use Relative L2 error (Li et al., 2021) for both training and evaluation in all of our experiments.

## 5.1 Comparison of Pretraining Strategies

We benchmark our proposed PDE pretraining strategies on different neural operators and datasets, and show the condensed results for auto-regressive prediction in Table 2. For a detailed comparison, we present results of different PDE pretraining strategies for fixed-future and auto-regressive tasks on all datasets in Appendix A. Additionally, we consider cases where the fine-tuning dataset contains coefficients unseen during pretraining, and present these out-of-distribution results in Appendix A as well.

Through these experiments, we find multiple insights. Firstly, we observe that the pretraining performance varies with the choice of model and dataset. Specifically, different models benefit differently from pretraining, as well as based on the predicted PDE and task (i.e. fixed-future vs. auto-regressive). However, transfer learning generally performs well across different tasks, models, and datasets, suggesting that it is a good choice for a pretraining task. This is also reflected in the literature, where previous work generally focuses on transferring knowledge between datasets (Chen et al., 2021; Goswami et al., 2022; Chakraborty et al., 2022; Tripura & Chakraborty, 2023) or pretrain by predicting physics of large datasets (Hao et al., 2024; McCabe et al., 2023; Subramanian et al., 2023). We hypothesize that transfer learning is effective since PDE data is inherently unlabeled; physics prediction uses future timesteps as a label, similar to next-token prediction for GPT models, which is cast as self-supervised learning. When the data is sufficient, using surrogate objectives such as derivatives or sorting sequences may not be as effective as the true objective of fixed-future of auto-regressive prediction. Another observation is that pretraining frameworks are generally dependent on specific architectures; for example, many CV pretraining strategies shuffle patches of data, which can introduce arbitrary discontinuities and high-frequency modes in FNO models, yet are not as challenging for convolutional models such as Unet. Furthermore, pretraining strategies are also dependent on the downstream task; for example, *Derivative* pretraining works well for auto-regressive prediction but not fixed future prediction, as the solution at a distant timestep is very different from the current derivatives, but the solution at the next timestep is highly dependent on the current derivatives.

Secondly, we observe that directly adapting computer vision methods to the physics domain generally results in poor performance. In many experiments, using a CV pretraining method would often hurt performance compared to not pretraining. This points to a general difference between CV and physics tasks. In the vision domain many downstream tasks are classification-based (i.e. ImageNet, Object Detection, etc.), which results in many pretraining tasks modeled around classification, whereas physics prediction is a high-dimensional regression task. Beyond this, physics predictions not only need to be visually consistent, but also numerically accurate, which can be difficult to learn from a classification task. In fact, using physics-based

Table 3: **Effects of Data Augmentation for Auto-regressive Prediction:** We present comparisons of different pretraining strategies combined with data augmentations after pretraining on the Heat, Advection, and Burgers equations and fine-tuning in 500 unseen samples. The data is distilled into two tables; for full results see Appendix B.

(a) **The best pretraining with data augmentation strategy varies with model and dataset.** Different models benefit from different augmentations when paired with pretraining strategies. Note: $^p$ denotes PICL and $^t$ denotes transfer learning.

| Model | Best Augmentation | | | |
|---|---|---|---|---|
| | Heat | Advection | Burgers | NS |
| FNO | Shift | Scale$^t$ | None$^p$ | None |
| DeepONet | Shift | Shift$^t$ | Shift$^t$ | Shift$^t$ |
| OFormer | Noise$^t$ | None$^p$ | Noise$^t$ | Noise$^t$ |
| Unet | Shift$^t$ | Shift$^p$ | Shift$^t$ | Noise |

(b) **Adding data augmentations consistently improves performance.** When choosing the correct combination of pretraining and data augmentation strategies, we find it improves performance during autoregressive prediction compared to baselines.

| Model | Improvement w/ Best Augmentation | | | |
|---|---|---|---|---|
| | Heat | Advection | Burgers | NS |
| FNO | 14.21% | 8.287% | 1.411% | 0.000% |
| DeepONet | 8.745% | 1.537% | 13.21% | 3.761% |
| OFormer | 35.35% | 4.426% | 16.288% | 13.23% |
| Unet | 30.051% | 0.735% | 10.322% | 3.434% |

pretraining methods, such as transferring between prediction tasks, regressing derivatives, or a physics-informed contrastive loss, generally results in better performance.

Lastly, we observe that different models have different capacities for pretraining. For example, the OFormer architecture, which is based on transformers, benefits greatly from pretraining in many scenarios; this could be because transformers lack inductive bias and can model arbitrary relationships. Furthermore, Unet architectures also benefit consistently from pretraining; this is reflected in common convolutional architectures used for pretraining in the CV domain, such as ResNet (He et al., 2015). DeepONet and FNO show smaller improvements with pretraining, suggesting that the architectures are less tailored for pretraining. This is especially true for FNO; we hypothesize that the learned Fourier modes may be very different between tasks, resulting in challenges when transferring weights to new tasks.

## 5.2 Comparison of Data Augmentations

To study the effects of augmenting data during pretraining and finetuning, we conduct experiments in which data augmentations are added to three pretraining strategies (*None, Transfer, PICL*). These experiments are run to compare data augmentations to a baseline model that is not pretrained, as well as its effects on the most effective pretraining strategies (i.e. *Transfer, PICL*). The results are summarized in Table 3 for auto-regressive prediction, and the complete results can be found in Appendix B. We find the best augmentation by considering the pretraining strategy and augmentation pairing with the lowest error. To calculate its improvement, this error is compared to a model that is not pretrained, however, to isolate the effect of pretraining, this can be compared to a corresponding pretrained model without data augmentation in the Appendix.

We find that different models benefit from different augmentations; for example, DeepONet performs well with shifted data, but OFormer performs well with noised data. However, across models, datasets, and downstream tasks, one can generally find a data augmentation that improves performance. This suggests that the most effective pretraining frameworks should incorporate a data augmentation strategy, and indeed the best-performing models considered in this study often make use of data augmentations. Furthermore, when an augmented model is compared to its baseline, the performance tends to increase even when the baseline is pretrained, albeit modestly. Transfer learning performs best in nine of our 12 cases, and shift augmentation performs best in eight of our 12 cases, with their combination performing best in six, suggesting that this combination improves performance best across different data sets and models. We believe that data augmentations can help due to the fact that PDE data remains scarce; numerical simulation is needed for high quality data, and as a result emulating a larger dataset with augmentations is beneficial.

Table 4: **Effects of Downstream Dataset:**  We compare the effect of pretraining without data augmentation when the downstream dataset is varied—both with the number of samples or the distribution of samples. We find that pretraining benefits more in data-scarce regimes, as well as when the downstream data is similar to the pretraining data.

(a) **Pretraining is more beneficial when downstream data is scarce.** Improvement is measured by comparing the best pretraining method with no pretraining. We report the average improvement across the Heat, Adv, and Burgers PDEs for a given # samples and model.

| # Samples | Best Improvement over None | | | |
| | FNO | DeepONet | OFormer | Unet |
|---|---|---|---|---|
| 100 | -5.953% | 6.755% | 47.60% | 23.05% |
| 250 | 4.111% | 7.361% | 29.60% | 18.18% |
| 500 | 6.875% | 6.630% | 19.57% | 13.59% |
| 1000 | 2.026% | 6.135% | 13.27% | 2.995% |

(b) **Pretraining improves performance for both in and out-of-distribution downstream tasks.** For each distribution of downstream data, we find the best improvement from pretraining and average across PDEs. Models show varying generalization capacities.

| Distribution | Best Improvement over None | | | |
| | FNO | DeepONet | OFormer | Unet |
|---|---|---|---|---|
| In | 6.875% | 6.630% | 19.57% | 13.592% |
| Out | 3.920% | -3.383% | 34.058% | 27.696% |
| NS | -8.658% | 2.899% | -1.608% | 1.865% |

## 5.3 Scaling Behavior

We compare the effect of pretraining for different numbers of downstream samples in Table 4. We measure this effect by finding the best pretraining method for a given model, PDE, and dataset size, then calculating its improvement over no pretraining; after calculating the improvement, we average this metric across the Heat, Advection, and Burgers PDEs for auto-regressive prediction. In general, we observe a trend in which the improvement of pretrained models diminishes as the number of fine-tuning samples increases, which is expected as fine-tuning data approaches the pretraining dataset size. It follows that if the downstream data is abundant, directly training on this would be optimal. Additionally, despite these trends, the relative improvement of different pretraining strategies remains approximately constant between different downstream dataset sizes. An exception to these trends is the FNO model; we hypothesize that learned Fourier modes may be more challenging to fine-tune than other learning mechanisms such as attention matrices or convolutional kernels.

For a detailed comparison of the scaling behavior in individual datasets and models, we refer readers to Appendix C. Empirically, we observe a higher variance between random seeds when using a smaller dataset for fine-tuning. Furthermore, the advection equation can generally be learned with fewer samples and the performance is approximately constant with increasing dataset size. Additionally, different models and pretraining strategies display different scaling behaviors, with some models and pretraining strategies displaying greater increases in performance when fine-tuning to scarce data. This further underscores the importance of proper architecture choices that scale well, such as using transformer-based neural operators. Lastly, scaling behavior is more pronounced in fixed-future experiments; this could be because there is less data in fixed-future experiments due to only predicting a single target per data sample as opposed to predicting multiple targets across a longer auto-regressive rollout.

## 5.4 Generalization Behavior

We compare the effect of varying the distribution of the downstream dataset on the performance of pretrained models. In particular, we compare fine-tuning to unseen coefficients of the same equation (*Out*) as well as fine-tuning to an unseen PDE with novel initial conditions and forcing terms (*NS*); these results are shown in Table 4 with 500 fine-tuning samples for auto-regressive prediction. In general, we observe reduced performance when fine-tuning to the Navier-Stokes equations, compared to fine-tuning to samples within the pretraining distribution (*In*). For certain models, this also holds when fine-tuning to a dataset with unseen coefficients (*Out*). These generalization behaviors are also approximately consistent between different sample sizes of the fine-tuning dataset. It is important to note that certain pretraining frameworks generalize better than others; for example, *Coefficient* pretraining largely hurts performance, since the fine-tuning distribution contains different coefficients by construction.

We note that the OFormer and Unet architectures show better performance when fine-tuning to out-of-distribution samples; we hypothesize that this is due to shifts in coefficients causing easier phenomena to model. For example, increasing the diffusivity in the heat equation causes transient effects to be concentrated in a few initial timesteps and sparse behavior for the majority of the rollout. Nevertheless, under certain conditions, pretraining shows generalization to unseen coefficients and PDEs, which is a promising direction.

## 6 Discussion

### 6.1 Theoretical Insights

There are many insights from the broader deep learning community that can help inform our results. In our study, an important factor in pretraining performance is the choice of model architecture, which is reflected both in both PDE and deep learning literature. Many recent PDE works have chosen to include attention modules in their architectures (Hao et al., 2024; Herde et al., 2024; Li et al., 2023a; McCabe et al., 2023; Hang et al., 2024), which is supported by the observation that transformer architectures can be expressive by making the least assumptions on the structure of data (Jiang et al., 2022). We similarly find that OFormer tends to display larger improvements compared to other architectures. Furthermore, Kolesnikov et al. (2020) find that training on larger datasets is vital for transferability, which we also observe by limiting the fine-tuning data.

Additionally, the use of data augmentations can be understood as assembling a more comprehensive training set such that the distance between the training set and validation/test sets is reduced (Shorten & Khoshgoftaar, 2019). In the context of PDEs, physics data can contain different coefficients, initial conditions, and boundary conditions; augmentations can help to mimic invariances to these conditions (e.g., spatially shifting data to model different initial conditions, scaling data to mimic different coefficients, etc.) such that models can perform well in spite of these challenges. The utility of data augmentation is further exacerbated by the difficulty in generating high-quality PDE data from numerical simulation, which could motivate future work in exploring more complex data augmentation approaches such as generating synthetic samples from GANs (Sandfort et al., 2019) or neural style transfer (Gatys et al., 2016).

Lastly, we consider the connection of different PDE pretraining strategies to broader deep learning pretraining insights. In general, it is understood that unsupervised learning is beneficial due to the abundance of unlabeled data and the human labor required to label samples (Bengio et al., 2021). However, in the context of PDE data, the goal is usually to predict a future frame of a current solution or to upsample low-resolution physics, both of which do not require human intervention to label data. In addition, numerical simulation is required to generate PDE data, which is the limiting factor in assembling large datasets. As a result, we hypothesize that this limitation is the main factor as to why unsupervised PDE methods relying on surrogate objectives do not perform as well as supervised pretraining, also known as transfer learning. Currently, unsupervised methods rely on the same data as supervised methods, both because of the computational cost of numerical simulation and because of the lack of human labor needed to label supervised samples. Addressing this limitation could be a promising direction for future work on unsupervised PDE learning.

### 6.2 Best Practices

Our experiments have demonstrated that transfer learning combined with the shift augmentation performs best in the majority of our experiments. For pretraining strategies, we believe that transfer learning performs best because additional pretraining strategies are not able to extract additional, task-specific information better than transfer learning. Having identical training tasks is simply more effective in this case. For data augmentation, the shift augmentation is the only augmentation that preserves the underlying dynamics of our data since all of our systems are spatially invariant. We believe that this improves performance over the other augmentation strategies, which do not preserve the underlying dynamics. For further applications, transfer learning + shift augmentation is a good initial pretraining strategy, but practitioners will likely need to tune and adapt additional strategies to achieve maximal performance. Furthermore, previous work (Brandstetter et al., 2022) has demonstrated improved performance by combining multiple augmentations, which may help across many systems. However, due to computational cost, we leave this to future work for

each practitioner's specific application, as best practices are likely model and dataset dependent. Lastly, to maximize the benefit of pretraining, we recommend collecting as much pretraining data as possible.

## 7 Conclusion

In this work, we compare pretraining strategies for PDEs by examining pretraining frameworks that can be used across different models and datasets. In particular, we consider adapting CV pretraining to the PDE domain through sorting spatio-temporal data to learn underlying dynamics without labels. Furthermore, we derive several PDE characteristics that can be predicted, such as its coefficients, derivatives, or reconstructed input. Lastly, we implement existing contrastive as well as transfer learning strategies to construct a diverse set of pretraining strategies. Notably, these strategies can be applied to any model and PDE problem and are flexible to future advances in architectures or datasets.

Through pretraining with different frameworks and data augmentations, we compare their effects on different PDEs, models, downstream datasets, and fine-tuning tasks. We find that pretraining can be highly dependent on model and dataset choices, but in general transfer learning or physics-based strategies do well. Furthermore, we find that directly adapting pretraining strategies from other domains often fails, motivating the need to design PDE-specific pretraining frameworks. Lastly, we observe that different models have different capacities for pretraining, with transformer and CNN based architectures benefiting the most from pretraining and highlighting the need for architectures that have high capacity and transferability.

To further understand PDE pretraining, we investigate the effect of adding data augmentations and varying the fine-tuning dataset. We find that data augmentations consistently benefit performance, with the shift augmentation showing best performance most often. Combining transfer learning with shift augmentation shows the best performance in the majority of test cases. Additionally, pretraining performance is accentuated when the fine-tuning dataset is scarce or similar to the pretraining distribution. Through establishing a deeper understanding of pretraining for PDEs, we hope that future work can leverage these insights to propose new pretraining strategies and expand on current architectures.

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

# A    Comparison of Pretraining Strategies

## A.1    Fixed Future Experiments

Table 5: Models are pretrained on 9216 combined 2D Heat, Advection, and Burgers samples and finetuned on 500 samples for each PDE. Normalized L2 errors ($\times 10^{-1}$) are calculated on 256 validation samples and averaged over five seeds. The lowest errors are given in dark grey , and second lowest errors are given in light grey.

(a) Fixed Future Pretraining Results.

| PDE | Model | None | Transfer | Binary | TimeSort | Jigsaw | Coefficient | Derivative | Masked | PICL |
|---|---|---|---|---|---|---|---|---|---|---|
| Heat | FNO | 0.240 | 0.467 | 0.813 | 0.771 | 0.838 | 0.387 | 2.742 | 0.557 | 0.182 |
| | DeepONet | 0.669 | 0.246 | 0.755 | 0.435 | 0.598 | 0.467 | 0.602 | 0.483 | 0.675 |
| | OFormer | 0.762 | 0.275 | 7.520 | 8.822 | 3.450 | 1.898 | 0.413 | 0.531 | 0.630 |
| | Unet | 0.150 | 0.061 | 0.378 | 0.339 | 0.483 | 0.234 | 0.131 | 0.118 | 0.145 |
| Adv | FNO | 3.533 | 1.517 | 7.741 | 5.427 | 6.138 | 6.442 | 6.205 | 4.522 | 3.555 |
| | DeepONet | 9.907 | 9.587 | 10.006 | 9.814 | 9.978 | 9.875 | 9.926 | 9.840 | 9.952 |
| | OFormer | 9.645 | 5.334 | 10.006 | 10.022 | 10.006 | 10.010 | 9.878 | 9.795 | 9.206 |
| | Unet | 3.962 | 1.488 | 5.747 | 5.568 | 6.509 | 9.286 | 4.909 | 4.058 | 3.802 |
| Burgers | FNO | 0.704 | 0.675 | 1.238 | 1.120 | 1.226 | 1.139 | 4.461 | 0.896 | 0.694 |
| | DeepONet | 4.096 | 3.37 | 4.758 | 3.638 | 3.869 | 3.776 | 3.987 | 3.674 | 4.195 |
| | OFormer | 1.92 | 1.517 | 9.318 | 9.792 | 5.024 | 3.610 | 2.112 | 1.997 | 1.994 |
| | Unet | 1.027 | 0.771 | 1.382 | 1.174 | 1.450 | 1.168 | 0.918 | 0.822 | 0.989 |
| NS | FNO | 2.112 | 2.147 | 3.500 | 6.285 | 3.530 | 3.874 | 6.200 | 2.386 | 2.232 |
| | DeepONet | 5.560 | 5.226 | 7.650 | 5.907 | 15.208 | 5.712 | 5.990 | 5.610 | 5.514 |
| | OFormer | 3.744 | 3.801 | 6.056 | 6.099 | 6.099 | 5.445 | 4.631 | 4.670 | 4.056 |
| | Unet | 2.279 | 1.403 | 3.261 | 2.847 | 3.488 | 2.493 | 2.341 | 2.332 | 2.262 |

(b) Out-of-Distribution Fixed Future Pretraining Results.

| PDE | Model | None | Transfer | Binary | TimeSort | Jigsaw | Coefficient | Derivative | Masked | PICL |
|---|---|---|---|---|---|---|---|---|---|---|
| Heat | FNO | 7.507 | 1.619 | 8.842 | 8.371 | 8.957 | 8.966 | 13.610 | 7.219 | 8.282 |
| | DeepONet | 1.008 | 3.187 | 1.507 | 2.570 | 13.584 | 2.845 | 1.846 | 1.517 | 2.089 |
| | OFormer | 11.142 | 6.87 | 12.291 | 11.981 | 12.106 | 11.568 | 11.862 | 11.408 | 11.317 |
| | Unet | 5.510 | 1.034 | 7.738 | 6.147 | 12.880 | 6.240 | 4.944 | 3.968 | 5.373 |
| Adv | FNO | 1.456 | 1.766 | 2.406 | 2.221 | 2.048 | 2.013 | 2.298 | 2.477 | 1.407 |
| | DeepONet | 9.600 | 9.558 | 9.955 | 9.581 | 9.654 | 9.616 | 9.558 | 9.555 | 9.563 |
| | OFormer | 8.749 | 8.23 | 9.923 | 9.974 | 9.750 | 10.013 | 7.558 | 9.187 | 8.775 |
| | Unet | 1.952 | 2.563 | 2.746 | 2.486 | 2.614 | 3.162 | 2.294 | 1.84 | 1.802 |
| Burgers | FNO | 0.195 | 0.454 | 0.685 | 0.688 | 0.742 | 0.307 | 1.734 | 0.627 | 0.19 |
| | DeepONet | 0.189 | 0.122 | 0.307 | 0.150 | 0.240 | 0.150 | 0.154 | 0.144 | 0.164 |
| | OFormer | 0.528 | 0.163 | 6.134 | 6.928 | 2.400 | 1.558 | 0.387 | 0.448 | 0.585 |
| | Unet | 0.342 | 0.054 | 0.333 | 0.218 | 0.333 | 0.147 | 0.202 | 0.240 | 0.340 |

## A.2 Auto-regressive Experiments

Table 6: Models are pretrained on 9216 combined 2D Heat, Advection, and Burgers samples and finetuned on 500 samples for each PDE. Normalized L2 errors ($\times 10^{-1}$) are calculated on 256 validation samples and averaged over five seeds. The lowest errors are given in dark grey, and second lowest errors are given in light grey.

(a) Autoregressive Pretraining Results.

| PDE | Model | None | Transfer | Binary | TimeSort | Jigsaw | Coefficient | Derivative | Masked | PICL |
|-----|-------|------|----------|--------|----------|--------|-------------|------------|--------|------|
| Heat | FNO | 2.730 | 5.507 | 3.888 | 4.704 | 3.878 | 4.838 | 2.336 | 3.984 | 2.584 |
| | DeepONet | 2.374 | 2.429 | 2.618 | 2.592 | 3.046 | 2.589 | 2.352 | 2.32 | 2.289 |
| | OFormer | 4.410 | 2.694 | 24.214 | 11.398 | 5.498 | 6.982 | 3.277 | 3.274 | 4.418 |
| | Unet | 3.357 | 2.378 | 3.286 | 2.768 | 2.586 | 2.406 | 2.464 | 2.39 | 3.019 |
| Adv | FNO | 30.890 | 28.586 | 30.669 | 29.888 | 31.571 | 29.571 | 29.875 | 29.584 | 30.676 |
| | DeepONet | 27.971 | 27.453 | 28.058 | 28.778 | 28.637 | 28.307 | 28.861 | 28.387 | 28.016 |
| | OFormer | 30.102 | 30.784 | 29.677 | 29.674 | 29.293 | 29.299 | 30.467 | 30.774 | 28.719 |
| | Unet | 30.640 | 30.832 | 30.17 | 30.058 | 30.992 | 31.027 | 30.579 | 30.310 | 30.355 |
| Burgers | FNO | 5.104 | 5.696 | 6.362 | 6.640 | 5.373 | 6.310 | 5.168 | 5.466 | 5.031 |
| | DeepONet | 5.101 | 4.298 | 5.706 | 5.341 | 5.638 | 5.702 | 5.024 | 5.190 | 5.167 |
| | OFormer | 7.734 | 6.41 | 25.731 | 18.102 | 10.157 | 10.026 | 7.059 | 7.101 | 8.293 |
| | Unet | 5.440 | 4.912 | 6.339 | 5.763 | 5.277 | 5.312 | 5.280 | 5.197 | 5.656 |
| NS | FNO | 5.884 | 6.708 | 8.626 | 11.211 | 7.293 | 7.276 | 9.245 | 6.393 | 6.086 |
| | DeepONet | 6.461 | 6.274 | 8.954 | 6.607 | 7.118 | 6.659 | 6.526 | 6.587 | 6.427 |
| | OFormer | 10.300 | 10.466 | 18.433 | 16.358 | 13.065 | 12.592 | 10.996 | 11.750 | 12.380 |
| | Unet | 5.854 | 5.745 | 6.461 | 6.110 | 6.233 | 6.011 | 5.902 | 5.813 | 6.285 |

(b) Out-of-Distribution Autoregressive Pretraining Results.

| PDE | Model | None | Transfer | Binary | TimeSort | Jigsaw | Coefficient | Derivative | Masked | PICL |
|-----|-------|------|----------|--------|----------|--------|-------------|------------|--------|------|
| Heat | FNO | 25.418 | 22.835 | 26.810 | 41.450 | 26.346 | 26.400 | 25.968 | 23.83 | 25.623 |
| | DeepONet | 3.062 | 3.914 | 3.981 | 4.208 | 15.344 | 5.101 | 4.291 | 29.002 | 3.166 |
| | OFormer | 35.888 | 17.203 | 33.862 | 34.864 | 35.389 | 33.472 | 32.803 | 34.528 | 32.857 |
| | Unet | 16.998 | 3.965 | 20.390 | 17.475 | 21.936 | 20.397 | 13.866 | 9.126 | 12.961 |
| Adv | FNO | 24.733 | 23.405 | 24.166 | 24.970 | 24.579 | 24.426 | 24.922 | 24.710 | 24.730 |
| | DeepONet | 26.480 | 26.179 | 27.219 | 26.755 | 26.832 | 26.864 | 26.214 | 26.288 | 26.406 |
| | OFormer | 25.210 | 25.344 | 29.968 | 28.877 | 25.325 | 25.658 | 25.174 | 24.269 | 25.014 |
| | Unet | 24.627 | 24.976 | 25.053 | 25.062 | 24.733 | 24.688 | 24.630 | 24.749 | 24.558 |
| Burgers | FNO | 1.443 | 3.830 | 2.688 | 4.630 | 2.778 | 2.646 | 1.498 | 3.376 | 1.424 |
| | DeepONet | 1.357 | 1.408 | 1.677 | 1.552 | 2.150 | 1.642 | 1.133 | 1.629 | 1.247 |
| | OFormer | 3.181 | 1.706 | 21.411 | 8.346 | 3.942 | 5.360 | 2.141 | 2.198 | 3.032 |
| | Unet | 1.594 | 1.491 | 1.930 | 1.706 | 1.661 | 1.517 | 1.514 | 1.498 | 1.804 |

# B    Comparison of Data Augmentations

## B.1    Fixed Future Experiments

Table 7: Models are pretrained on 9216 combined 2D Heat, Advection, and Burgers samples and finetuned on 500 samples for each PDE. Each baseline (e.g, None, PICL, Transfer) is followed by its variants with different augmentations (e.g, Noise, P-Shift, T-Scale). Normalized L2 errors ($\times 10^{-1}$) are calculated on 256 validation samples and averaged over five seeds. The lowest errors are given in dark grey , and second lowest errors are given in light grey.

(a) Fixed Future Pretraining Results.

| PDE | Model | None | Noise | Shift | Scale | PICL | P-Noise | P-Shift | P-Scale | Transfer | T-Noise | T-Shift | T-Scale |
|---|---|---|---|---|---|---|---|---|---|---|---|---|---|
| Heat | FNO | 0.246 | 0.183 | 0.182 | 0.179 | 0.182 | 0.389 | 0.332 | 0.169 | 0.418 | 0.407 | 0.445 | 0.449 |
| | DeepONet | 0.670 | 0.638 | 0.637 | 0.661 | 0.493 | 0.455 | 0.457 | 0.562 | 0.449 | 0.407 | 0.401 | 0.434 |
| | OFormer | 0.763 | 0.482 | 0.535 | 0.650 | 0.901 | 0.425 | 0.467 | 0.918 | 0.510 | 0.340 | 0.338 | 0.497 |
| | Unet | 0.147 | 0.163 | 0.162 | 0.092 | 0.145 | 0.176 | 0.099 | 0.163 | 0.130 | 0.137 | 0.133 | 0.081 |
| Adv | FNO | 3.539 | 3.509 | 3.549 | 3.339 | 3.631 | 3.722 | 3.704 | 3.464 | 2.017 | 1.703 | 1.679 | 1.695 |
| | DeepONet | 9.906 | 9.807 | 9.792 | 9.794 | 9.713 | 9.594 | 9.599 | 9.737 | 9.606 | 9.565 | 9.557 | 9.560 |
| | OFormer | 9.643 | 9.508 | 9.566 | 8.661 | 9.690 | 9.689 | 9.591 | 9.559 | 9.296 | 8.818 | 8.716 | 6.948 |
| | Unet | 3.979 | 3.893 | 3.906 | 3.457 | 3.802 | 3.320 | 3.142 | 3.473 | 2.401 | 2.211 | 2.192 | 1.930 |
| Burgers | FNO | 0.698 | 0.583 | 0.584 | 0.721 | 0.640 | 0.643 | 0.627 | 0.761 | 0.645 | 0.643 | 0.636 | 0.831 |
| | DeepONet | 4.092 | 3.840 | 3.841 | 3.791 | 4.101 | 3.929 | 3.937 | 4.018 | 3.956 | 3.774 | 3.770 | 3.854 |
| | OFormer | 1.919 | 1.612 | 1.617 | 1.811 | 2.408 | 1.718 | 1.778 | 2.158 | 1.752 | 1.496 | 1.506 | 1.556 |
| | Unet | 1.026 | 0.950 | 0.988 | 0.844 | 0.989 | 0.722 | 0.633 | 0.727 | 0.952 | 0.877 | 0.883 | 0.812 |
| NS | FNO | 2.112 | 2.122 | 2.124 | 2.301 | 2.232 | 2.574 | 2.534 | 2.335 | 2.428 | 2.581 | 2.602 | 2.747 |
| | DeepONet | 5.560 | 5.543 | 5.544 | 5.552 | 5.514 | 5.316 | 5.320 | 5.517 | 5.492 | 5.313 | 5.313 | 5.333 |
| | OFormer | 3.744 | 3.415 | 3.399 | 3.569 | 4.056 | 3.696 | 3.694 | 3.922 | 3.962 | 3.629 | 3.609 | 3.605 |
| | Unet | 2.279 | 2.247 | 2.238 | 2.309 | 2.262 | 2.290 | 2.131 | 2.318 | 2.678 | 2.603 | 2.573 | 2.520 |

(b) Out-of-Distribution Fixed Future Pretraining Results.

| PDE | Model | None | Noise | Shift | Scale | PICL | P-Noise | P-Shift | P-Scale | Transfer | T-Noise | T-Shift | T-Scale |
|---|---|---|---|---|---|---|---|---|---|---|---|---|---|
| Heat | FNO | 7.721 | 7.818 | 7.805 | 6.409 | 8.282 | 8.670 | 8.807 | 7.528 | 2.431 | 2.842 | 2.726 | 2.806 |
| | DeepONet | 1.019 | 1.056 | 1.073 | 1.027 | 2.089 | 2.023 | 2.091 | 1.228 | 1.150 | 1.368 | 1.384 | 1.250 |
| | OFormer | 11.14 | 11.58 | 11.61 | 11.78 | 11.32 | 11.34 | 11.42 | 11.51 | 11.46 | 11.00 | 11.18 | 9.956 |
| | Unet | 5.504 | 5.128 | 5.203 | 3.299 | 5.373 | 5.070 | 3.894 | 3.261 | 2.491 | 2.139 | 2.040 | 1.783 |
| Adv | FNO | 1.457 | 1.374 | 1.372 | 1.305 | 1.407 | 1.496 | 1.487 | 1.317 | 1.583 | 1.601 | 1.604 | 1.537 |
| | DeepONet | 9.599 | 9.581 | 9.585 | 9.584 | 9.563 | 9.513 | 9.497 | 9.523 | 9.511 | 9.511 | 9.493 | 9.492 |
| | OFormer | 8.750 | 8.214 | 8.251 | 7.179 | 8.775 | 8.833 | 8.682 | 8.472 | 8.997 | 9.082 | 8.968 | 7.807 |
| | Unet | 1.955 | 1.967 | 2.014 | 1.623 | 1.802 | 1.597 | 1.582 | 1.281 | 2.117 | 2.137 | 2.165 | 1.757 |
| Burgers | FNO | 0.214 | 0.173 | 0.170 | 0.148 | 0.190 | 0.261 | 0.265 | 0.115 | 0.400 | 0.425 | 0.455 | 0.592 |
| | DeepONet | 0.188 | 0.145 | 0.146 | 0.182 | 0.164 | 0.126 | 0.131 | 0.192 | 0.126 | 0.122 | 0.122 | 0.122 |
| | OFormer | 0.526 | 0.288 | 0.326 | 0.536 | 0.585 | 0.289 | 0.343 | 0.780 | 0.362 | 0.201 | 0.201 | 0.368 |
| | Unet | 0.341 | 0.322 | 0.323 | 0.313 | 0.340 | 0.281 | 0.216 | 0.303 | 0.284 | 0.229 | 0.223 | 0.252 |

## B.2 Auto-regressive Results

Table 8: Models are pretrained on 9216 combined 2D Heat, Advection, and Burgers samples and finetuned on 500 samples for each PDE. Each baseline (e.g, None, PICL, Transfer) is followed by its variants with different augmentations (e.g, Noise, P-Shift, T-Scale). Normalized L2 errors ($\times 10^{-1}$) are calculated on 256 validation samples and averaged over five seeds. The lowest errors are given in dark grey, and second lowest errors are given in light grey.

(a) Autoregressive Pretraining Results.

| PDE | Model | None | Noise | Shift | Scale | PICL | P-Noise | P-Shift | P-Scale | Transfer | T-Noise | T-Shift | T-Scale |
|---|---|---|---|---|---|---|---|---|---|---|---|---|---|
| Heat | FNO | 2.731 | 2.345 | 2.343 | 2.405 | 2.584 | 2.908 | 2.830 | 2.428 | 4.338 | 5.398 | 5.174 | 5.215 |
| | DeepONet | 2.184 | 2.410 | 1.993 | 2.158 | 2.289 | 2.120 | 2.166 | 2.371 | 2.347 | 2.297 | 2.118 | 2.378 |
| | OFormer | 4.540 | 3.778 | 3.854 | 4.074 | 4.418 | 3.658 | 3.433 | 4.674 | 3.190 | 2.935 | 3.010 | 3.218 |
| | Unet | 3.301 | 2.695 | 2.776 | 2.387 | 3.019 | 2.339 | 2.349 | 2.494 | 2.301 | 2.340 | 2.276 | 2.494 |
| Adv | FNO | 30.89 | 30.88 | 31.02 | 30.67 | 30.68 | 29.94 | 30.22 | 30.11 | 28.59 | 28.33 | 28.65 | 28.33 |
| | DeepONet | 27.98 | 28.66 | 28.04 | 28.01 | 28.02 | 28.13 | 27.840 | 28.20 | 28.14 | 28.04 | 27.55 | 27.81 |
| | OFormer | 30.05 | 31.04 | 30.29 | 30.31 | 28.72 | 28.89 | 28.84 | 29.83 | 30.18 | 30.23 | 30.61 | 30.11 |
| | Unet | 29.94 | 30.56 | 30.12 | 30.55 | 30.36 | 30.37 | 29.72 | 30.64 | 30.45 | 30.17 | 30.64 | 30.75 |
| Burgers | FNO | 5.103 | 5.127 | 5.076 | 5.302 | 5.031 | 5.098 | 5.142 | 5.311 | 5.874 | 6.428 | 5.988 | 7.448 |
| | DeepONet | 4.884 | 5.080 | 4.740 | 4.577 | 5.167 | 4.675 | 4.666 | 4.671 | 4.757 | 4.484 | 4.239 | 4.653 |
| | OFormer | 7.754 | 7.157 | 7.058 | 7.519 | 8.293 | 7.267 | 7.044 | 8.183 | 6.867 | 6.491 | 6.538 | 6.866 |
| | Unet | 5.503 | 5.334 | 5.335 | 5.202 | 5.656 | 5.206 | 5.229 | 4.980 | 4.957 | 4.976 | 4.935 | 4.940 |
| NS | FNO | 5.884 | 6.129 | 6.092 | 6.032 | 6.086 | 6.068 | 6.078 | 5.973 | 6.724 | 6.874 | 6.931 | 7.553 |
| | DeepONet | 6.461 | 6.397 | 6.404 | 6.400 | 6.427 | 6.393 | 6.376 | 6.434 | 6.310 | 6.241 | 6.218 | 6.276 |
| | OFormer | 10.30 | 8.96 | 9.011 | 9.012 | 12.38 | 10.60 | 10.25 | 11.16 | 10.78 | 8.937 | 9.037 | 9.183 |
| | Unet | 5.854 | 5.653 | 5.799 | 5.851 | 6.285 | 5.880 | 5.883 | 5.676 | 5.756 | 5.686 | 5.705 | 5.697 |

(b) Out-of-Distribution Autoregressive Pretraining Results.

| PDE | Model | None | Noise | Shift | Scale | PICL | P-Noise | P-Shift | P-Scale | Transfer | T-Noise | T-Shift | T-Scale |
|---|---|---|---|---|---|---|---|---|---|---|---|---|---|
| Heat | FNO | 25.42 | 25.48 | 25.50 | 23.92 | 25.62 | 26.01 | 25.84 | 23.69 | 23.89 | 25.775 | 25.82 | 25.73 |
| | DeepONet | 3.267 | 3.182 | 3.129 | 3.197 | 3.166 | 3.400 | 3.232 | 3.119 | 3.704 | 4.027 | 3.922 | 3.445 |
| | OFormer | 35.67 | 36.71 | 36.93 | 35.54 | 32.86 | 32.47 | 32.53 | 35.80 | 25.34 | 26.61 | 25.97 | 23.23 |
| | Unet | 16.95 | 15.37 | 15.66 | 11.66 | 12.96 | 9.51 | 10.43 | 8.279 | 4.649 | 4.563 | 4.687 | 4.172 |
| Adv | FNO | 24.73 | 25.06 | 25.05 | 24.92 | 24.73 | 25.22 | 25.16 | 24.96 | 23.69 | 23.39 | 23.15 | 23.316 |
| | DeepONet | 26.48 | 26.03 | 25.81 | 25.64 | 26.41 | 25.96 | 25.61 | 25.64 | 26.23 | 25.74 | 25.98 | 25.57 |
| | OFormer | 25.24 | 25.27 | 25.36 | 25.08 | 25.01 | 24.69 | 24.69 | 25.35 | 25.26 | 25.36 | 25.23 | 24.99 |
| | Unet | 24.48 | 24.96 | 24.70 | 24.90 | 24.56 | 24.93 | 24.58 | 24.81 | 25.14 | 24.46 | 24.88 | 25.09 |
| Burgers | FNO | 1.442 | 1.414 | 1.443 | 1.469 | 1.424 | 2.121 | 1.787 | 1.597 | 4.185 | 4.580 | 4.505 | 5.062 |
| | DeepONet | 1.212 | 1.335 | 1.167 | 1.229 | 1.247 | 1.273 | 1.261 | 1.291 | 1.411 | 1.278 | 1.467 | 1.460 |
| | OFormer | 3.135 | 2.624 | 2.688 | 2.920 | 3.032 | 2.437 | 2.216 | 3.404 | 2.044 | 1.941 | 1.908 | 2.161 |
| | Unet | 1.560 | 1.471 | 1.543 | 1.632 | 1.804 | 1.529 | 1.637 | 1.463 | 1.377 | 1.348 | 1.477 | 1.447 |

## C   Comparison of Downstream Dataset Size

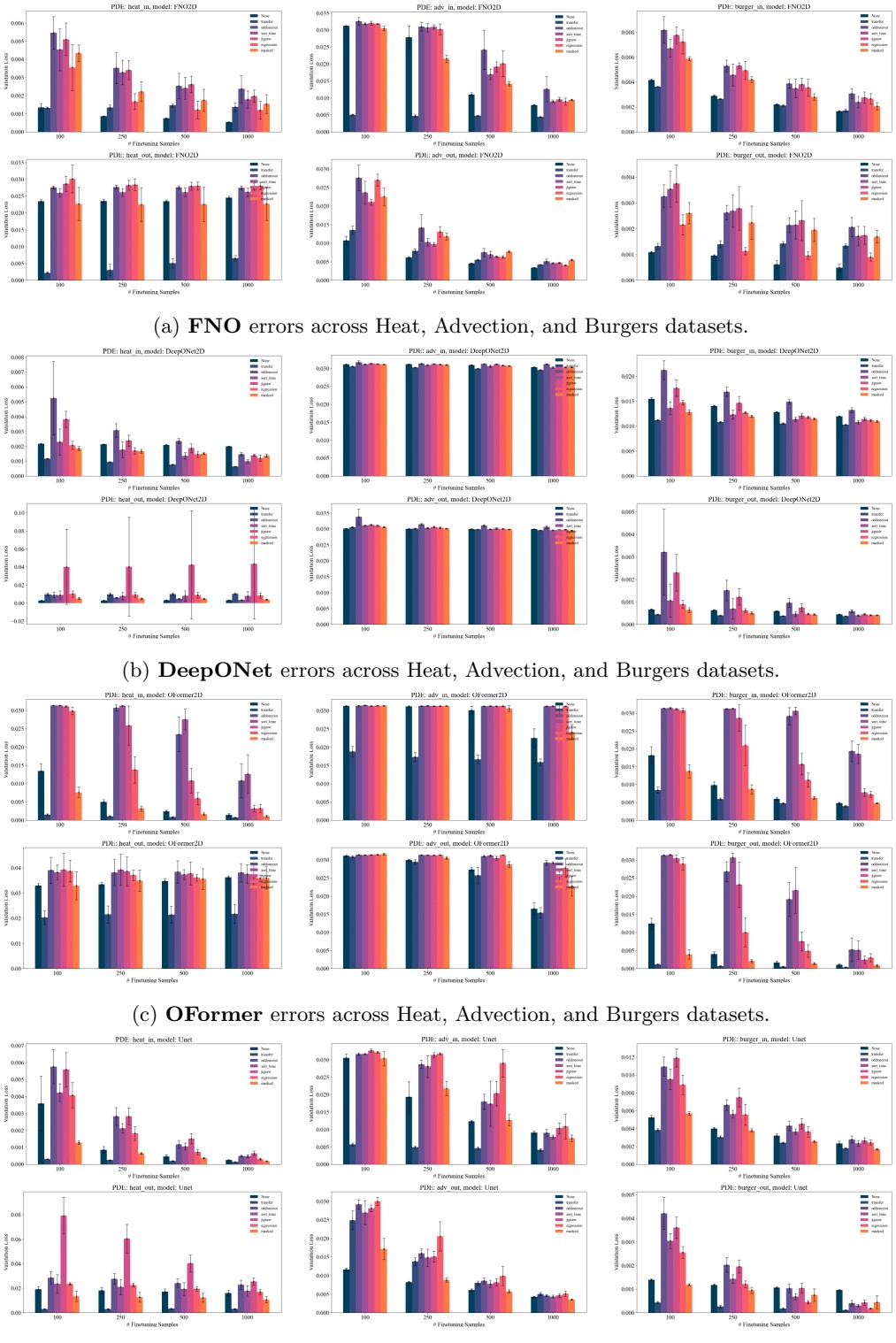

(a) **FNO** errors across Heat, Advection, and Burgers datasets.

(b) **DeepONet** errors across Heat, Advection, and Burgers datasets.

(c) **OFormer** errors across Heat, Advection, and Burgers datasets.

(d) **Unet** errors across Heat, Advection, and Burgers datasets.

Figure 3: **Fixed Future Scaling Behavior:** For each model, a specific PDE/distribution is displayed. Within each graph, the error of various pretraining strategies at different sample sizes is displayed. Validation errors are averaged over 5 seeds, and error bars denote 1 std-dev. Derivative errors are omitted as outliers.

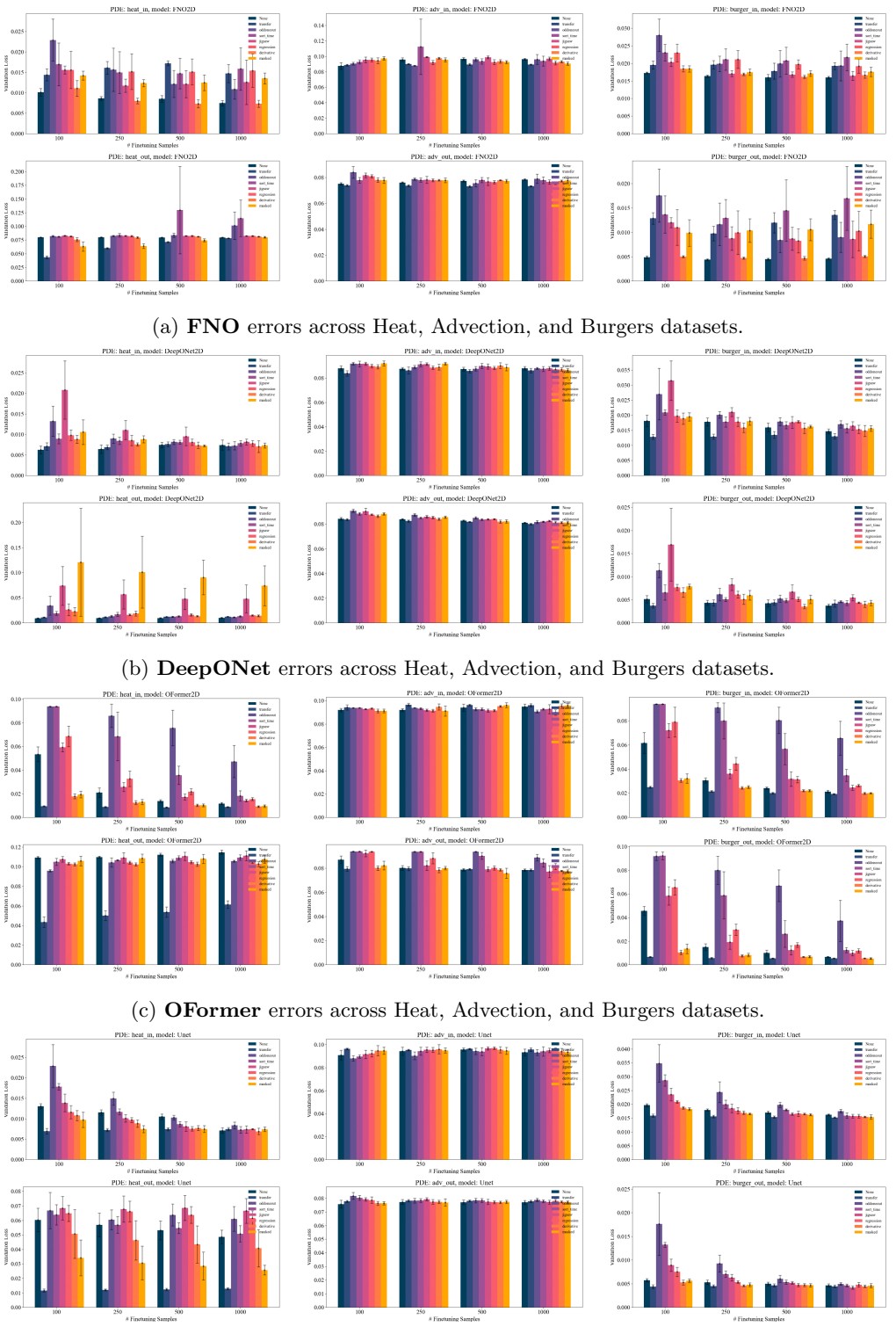

(a) **FNO** errors across Heat, Advection, and Burgers datasets.

(b) **DeepONet** errors across Heat, Advection, and Burgers datasets.

(c) **OFormer** errors across Heat, Advection, and Burgers datasets.

(d) **Unet** errors across Heat, Advection, and Burgers datasets.

Figure 4: **Auto-regressive Scaling Behavior:** For each model, a specific PDE/distribution is displayed. Within each graph, the performance of various pretraining strategies at different sample sizes is displayed. Validation errors are averaged over 5 seeds, and error bars denote 1 std-dev.

# D Implementation Details

## D.1 Dataset Details

We generate data according to the equations outlined in 4.1. We provide additional details here:

**Pretraining** During pretraining, 9216 total samples are generated, with 3072 samples of the 2D Heat, Advection, and Burgers equations respectively. The samples are generated with a resolution of $(n_t, n_x, n_y) = (32, 64, 64)$ or $(n_t, n_x, n_y) = (32, 32, 32)$ on the domain $(x, y) = [-1, 1]^2$ from $t = 0$ to $t = 2$; the discretization depends on the downstream resolution of the data. We sample equation coefficients from a defined pretraining distribution. Heat, Advection, and Burgers equation samples are generated with a finite-differences scheme; a first-order central difference is used to discretize the diffusive term, a first-order upwinding scheme is used to discretize the nonlinear convection term, and time is discretized with a forward Euler scheme. In addition, the advection equation is solved with its analytical solution.

**Training/Finetuning** During training/fine-tuning, we generate equations using a procedure similar to pretraining and sample coefficients either in the pretraining distribution or from a disjoint distribution to test generalization to unseen coefficients. For fine-tuning on the Navier-Stokes equations, we use a higher resolution of $(n_t, n_x, n_y) = (32, 64, 64)$, otherwise experiments are run with a resolution of $(n_t, n_x, n_y) = (32, 32, 32)$. We generate 1024 samples for the Heat, Advection, Burgers, and Navier-Stokes equations to train with. An additional 1024 out-of-distribution samples for the Heat, Advection, and Burgers equations is also generated. Additionally, the Burgers equation, initial conditions are unchanged to evaluate fine-tuning to a reference problem undergoing different dynamics, such as in design optimization problems (Cheng et al., 2024).

**Validation** Validation samples are generated similarly to fine-tuning samples, also with equation coefficients sampled from either the pretraining or disjoint distribution. We generate 256 samples for the Heat, Advection, Burgers, and Navier-Stokes equations.

## D.2 Model Details

Table 9: Hyperparameters for architectures used.

(a) FNO

| Parameter | Value |
|---|---|
| Modes | 4 |
| Width | 48 |
| # Layers | 4 |
| # Params | 300k |

(b) DeepONet

| Parameter | Value |
|---|---|
| Branch Size | 256 |
| Trunk Size | 256 |
| Branch Layers | 3 |
| Trunk Layers | 3 |
| Activation | SiLU |
| # Params | 250k |

(c) OFormer

| Parameter | Value |
|---|---|
| Hidden dim | 32 |
| Heads | 2 |
| Encoder depth | 2 |
| Decoder depth | 1 |
| Latent channels | 32 |
| # Params | 70k |

(d) Unet

| Parameter | Value |
|---|---|
| Hidden channels | 16 |
| # Blocks | 8 |
| Dim Scaling | (1,2,4) |
| # Params | 1M |

We implement modern FNO and Unet architectures according to Gupta & Brandstetter (2022). Furthermore, we implement DeepONet architectures according to DeepXDE (Lu et al., 2021b), and use the original implementation for OFormer (Li et al., 2023a). The hyperparameters used for the models are described in Table 9. For FNO, we use the standard spectral convolution and residual layers, as well as additional input channels for the x, y, and time coordinates and multiple timesteps. The Unet implementation uses multiple ConvNext blocks followed by an upsample/downsample block at each layer. Additionally, an attention block is used at the middle layer between upsampling and downsampling. When implementing OFormer, an attention module is used to aggregate information from the inputs as well as output query coordinates; this is aggregated into a latent encoding which is propagated with an MLP and decoded at future timesteps. Lastly, DeepONet uses an implementation that bundles multiple timesteps into both the branch and trunk net MLPs, these predictions are combined to ouput multiple future timesteps.

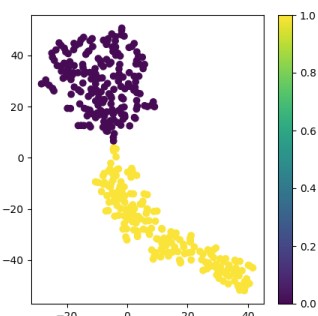 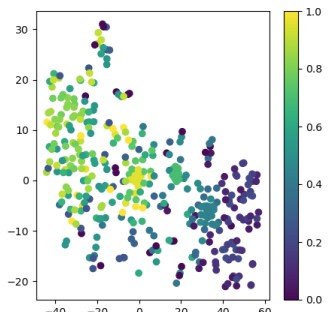 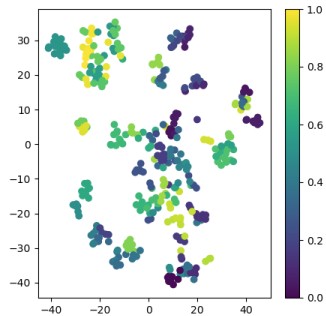

(a) Embeddings after *Binary* pre-training. Labels are defined as 0 for shuffled or 1 for original samples.

(b) Embeddings after *TimeSort* pretraining. Labels are scaled between 0 and 1 for 24 classes.

(c) Embeddings after *Jigsaw* pre-training. Labels are scaled between 0 and 1 for 1000 classes.

Figure 5: **t-SNE Embeddings of CV Pretrained Models:** We display latent embeddings after pretraining models on *Binary*, *TimeSort.* or *Jigsaw* objectives. We see that models learn to sort/classify shuffled samples well and can visualize the relative difficulties of the proposed pretraining strategies.

### D.3 Pretraining Details

During pretraining, different strategies require different implementations and hyperparameters. A consideration is that many models need a linear head during pretraining to project model outputs to the classification or regression dimension. Since models will be used for physics prediction, their outputs will be in the shape of the solution field, rather than cross entropy probabilities or regressed values. We use a lightweight CNN projector to downsample and flatten model outputs to the desired dimension. Models are generally trained for 200 epochs with a batch size of 32 using Adam with a learning rate of 1e-3, a weight decay of 1e-6, and a OneCycle scheduler for five seeds. A note is that for binary pretraining, the task is much easier and therefore the pretraining can be done with 100 epochs.

**Binary:** Binary pretraining is implemented by shuffling a sample in time and randomly choosing a shuffled or original input with corresponding labels of 0 or 1 to be used for classification. We use a CNN head to project model outputs to a single logit for a binary cross-entropy loss. Within this framework there are a few design decisions. The difficulty of the task can be modulated by the Hamming distance between the shuffled sample and the original sample. For example, if the shuffled sample is not changed much (e.g. only two frames are swapped), the difference between a sorted and shuffled sample is small and thus more challenging to distinguish. We can leverage this to gradually decrease the Hamming distance of shuffled samples to incrementally increase the difficulty of the task over pretraining. Empirically, this does not make a large difference during training so we choose to omit this curriculum learning for simplicity.

An additional consideration is the probability of sampling a shuffled or sorted sample. In theory, there are many more shuffled samples than sorted samples (i.e. more labels with 0 vs. 1); therefore, it may be beneficial to sample more shuffled samples and use a weighted binary cross-entropy loss. In practice this does not significantly affect training, so we uniformly sample sorted or shuffled samples. A final consideration is that PDE solutions generally do not exhibit large changes in time, therefore, we patchify the time dimension when shuffling to create larger changes in shuffled patches. In general, models are able to learn to distinguish between shuffled and original inputs very well, and we display t-SNE embeddings of a pretrained FNO model on a validation set of shuffled and unshuffled samples in Figure 5a.

**TimeSort/SpaceSort:** Sorting along a single dimension is implemented by patchifying the solution field along the desired dimension and shuffling these patches. This is done to create more distinct differences in the shuffled solution, with the patch size controlling the number of permutations of the

shuffled sequence. The permutation number affects the difficulty of the sorting task, with large permutation numbers being more difficult since each permutation represents a different class. To mitigate this, we set the patch size to ensure a sequence length of 4 when shuffling, resulting in $4! = 24$ classes or permutations of the solution field. The CNN projection head is modified accordingly to output 24 logits for a cross-entropy loss. In general, spatial sorting does not work well nor does training converge, so we omit this from the results; aliasing effects or periodic boundary conditions can make some spatially shuffled samples extremely similar or identical to sorted samples. However, temporal sorting tends to work well, and we display t-SNE embeddings of a pretrained FNO model on a validation set of temporally shuffled samples in Figure 5b.

**Jigsaw:** Jigsaw is implemented similarly to other sorting frameworks, however due to sorting along multiple axes the number of possible shuffled sequences quickly increases. We mitigate this by using spatial and temporal patches to ensure a sequence length of 8 when shuffled, resulting in $8! = 40320$ possible permutations. This is still a large number of classes for a task, therefore we deterministically choose 1000 samples with the largest Hamming distance between the shuffled sequence and original sequence. Contrary to the binary case, shuffled samples with larger Hamming distances are more challenging due to needing to sort more patches. The CNN projection head is modified accordingly to output 1000 logits for a cross-entropy loss. In general, jigsaw sorting tends to be more challenging, however, models can still display reasonable performance; we display t-SNE embeddings of a pretrained FNO model on a validation set of jigsaw shuffled samples in Figure 5c.

**Coefficient:** Coefficient regression is implemented by extracting coefficient values from the PDE metadata. The CNN projection head is then modified to output the corresponding number of logits for an MSE loss.

**Derivative:** We generate labels for derivative regression through taking spatial and time derivatives $\{u_t, u_x, u_y, u_{xx}, u_{yy}\}$ of the PDE solution field using FinDiff (Baer, 2018). This introduces an additional design consideration as the label has more values than the input. We modify the CNN projection head to upsample model outputs after convolution to the desired dimension and apply an MSE loss.

**Masked:** Masked inputs are generated by splitting inputs into spatial and temporal patches, and selecting a random subset of these to be masked. In our experiments, we choose to mask 75% of patches. Masked patches are replaced with a learnable mask token, and the full input is passed to the model to reconstruct the original solution field. Since the output shape is the same as the downstream target, a projection head is not strictly needed, but we still include a CNN projection head and apply an MSE loss. This follows previous work; models learn transferable latent features by abstracting reconstruction-specific behavior to a decoder (Chen et al., 2020; He et al., 2021).

**PICL:** PICL uses the Generalized Contrastive Loss function (Leyva-Vallina et al., 2023) given in equation 7:
$$\mathcal{L}_{GCL}(u_i, u_j) = \frac{\psi_{i,j}}{2} d_{physics}(u_i, u_j)^2 + \frac{1 - \psi_{i,j}}{2} \max(\tau - d_{physics}(u_i, u_j), 0)^2 \tag{7}$$

When working with multiple data sets simultaneously, a vector of operator coefficients is constructed as $\theta$. The similarity between systems is given by magnitude-aware cosine similarity: $\psi_{i,j}(\theta_i, \theta_j) = \frac{\sqrt{|\theta_i \cdot \theta_j|}}{\max(\|\theta_i\|, \|\theta_j\|)}$. The distance between samples is calculated in two parts for a given time $t$: $d_{system}(u_i, u_j) = u_i^{t+1} - u_j^t$, and $d_{update} = F(G_\Theta(u_i)) - G_\Theta(u_j)$, where $G_\Theta$ is our parameterized model, and $F(\cdot)$ is our numerical update. $d_{update}$ is anchored to $d_{system}$ to account for mode collapse, giving us the loss function: $d_{physics}(u_i, u_j) = \|d_{system}(u_i, u_j) - d_{update}(u_i, u_j)\|^2$. $\tau$ is a hyperparameter that defines a margin, above which samples are considered to be from different classes. For pretraining, we construct the operator coefficient vector as $\theta = [\|\mathbf{c}_{Burgers}\|, \nu, \|\mathbf{c}_{Advection}\|]$

## D.4 Data Augmentation Details

We implement three data augmentations to evaluate their effects on model performance: noise, shift, and scale.

**Noise** Gaussian noise is added to data samples and targets through sampling a Gaussian at zero mean and a prescribed variance: $X_{noise} = X + \sigma^2 \mathcal{N}(0, I)$. Empirically, we set the variance to $10^{-7}$; when noise levels are too high, model performance can significantly deteriorate.

**Shift** Using the Fourier shift theorem, samples can be shifted in space and resampled in the spectral domain (Brandstetter et al., 2022). Shifting PDE solutions in space preserves physics, since the PDEs considered in this work are invariant across space. Mathematically, this can be verified by deriving or looking up the Lie groups for the 2D Advection, Heat, and Burgers equations, for which there are many, and noting that the solutions can be shifted along the X or Y axes (Ibragimov, 1993). We uniformly sample the magnitude of the shift between $[-0.5, 0.5]$.

**Scale** Scaling PDE solutions respects physics for the Heat and Advection equations, but not the Burgers equation. However, we still choose to include this augmentation to evaluate the effect of physically inconsistent augmentations; in practice, scaling PDE solutions still improves model performance. The implementation is done by multiplying PDE solutions by a constant, which we uniformly sample between $[-0.5, 0.5]$.

### D.5 Fine-tuning Details

During fine-tuning, models trained until convergence for fixed-future or auto-regressive prediction and repeated for five seeds. In fixed-future prediction, models are given the solution field at $t = [0, 8)$ and the target is at $t = 32$. For auto-regressive prediction, models are given the solution field at $t = [0, 8)$ and the target is at $t = [8, 16)$. After this prediction, the models use their own output to predict the next step $t = [16, 24)$ until the time horizon of $t = 32$. To stabilize auto-regressive rollout, we implement temporal bundling and the pushforward trick (Brandstetter et al., 2023). Losses are calculated using a relative L2 norm (Li et al., 2021); validation losses are averaged across batch size and accumulated over timesteps or, in the case of fixed-future prediction, at only one timestep. For experiments with different fine-tuning sample sizes, samples are randomly chosen from 1024 possible samples to reach the desired number of samples for each seed. These fine-tuning datasets are independently resampled from the same pool for each seed, and larger datasets are not required to be strict supersets of the smaller datasets, which could increase the variance of results. We use an Adam optimizer with a learning rate of 1e-3, with weight decay of 1e-6, and a CosineAnnealing scheduler. All experiments are run on a NVIDIA GeForce RTX 2080Ti GPU.

# E  Auxiliary Results

During our experimentation, there were different pretraining strategies that did not improve model performance over a baseline of no pretraining. These are omitted from consideration in the results and analysis, however, for completeness, preliminary results that informed this decision are reported here.

## E.1  Sorting Spatially Shuffled Sequences

In this section, we compare SpaceSort with a baseline without pretraining. Both models are fine-tuned on 500 samples that are within the pretraining distribution from the respective PDEs on a fixed-future prediction objective. We hypothesize that spatially shuffling PDEs may confuse models during pretraining since it destroys important information, as physical systems need to be continuous in space.

Table 10: Models are pretrained on 9216 combined 2D Heat, Advection, and Burgers samples and finetuned on 500 samples for each PDE. L2 errors ($\times 10^{-3}$) are calculated on 256 validation samples and averaged over five seeds.

Table 11: Fixed-Future Pretraining Results with SpaceSort

| PDE | Model | None | SpaceSort |
|---|---|---|---|
| Heat | FNO | 0.75 | 2.61 |
| | DeepONet | 2.13 | 2.39 |
| | OFormer | 2.38 | 10.78 |
| | Unet | 0.46 | 1.51 |
| Adv | FNO | 11.04 | 19.17 |
| | DeepONet | 30.95 | 31.17 |
| | OFormer | 30.13 | 31.26 |
| | Unet | 12.37 | 20.34 |
| Burgers | FNO | 2.20 | 3.83 |
| | DeepONet | 14.04 | 14.63 |
| | OFormer | 5.99 | 15.70 |
| | Unet | 3.21 | 4.53 |

## E.2  Lie Contrastive Self-Supervised Learning

In this section, we compare VICRegBardes et al. (2022) against baseline FNO training for a fixed-future prediction task. We use values of $\nu = 1$, $\mu = 25$, $\lambda = 25$, matching values from the original paper. In this case, we use the first 8 frames to predict the system state at frame 32.

Table 12: Models are pretrained on 9216 combined 2D Heat, Advection, and Burgers samples and finetuned on 100 samples from the Heat equation. L2 errores ($\times 10^3$) are calculated on 256 validation samples and averaged over five seeds.

Table 13: Fixed-Future Pretraining Results with VICReg

| Dataset | FNO | FNO VICReg |
|---|---|---|
| Heat | 1.45 | 8.03 |

