# OpenReview forum: "Strategies for Pretraining Neural Operators"
_TMLR — Accepted by TMLR_

### Review · Reviewer_yf78 · 2024-07-09

**Summary Of Contributions:**

The authors compared pretraining and data augmentation methods for PDE models and attempted to adapt pretraining strategies in computer vision to PDE. From experimental results, the authors draw the following conclusions. First, the best-performing pretraining or augmentation method varies for different network architectures and problems. Second, transfer learning and shift augmentation generally perform best. Third, directly adapting CV methods do not work well. Fourth, pretraining is more effective when finetuning is performed on smaller datasets and similar data distributions.

**Audience:**

Yes

**Broader Impact Concerns:**

None.

**Claims And Evidence:**

No

**Requested Changes:**

I would suggest the authors consider the following changes to improve paper.

- Elaborate on the motivation for adapting CV pretraining methods to PDE.
- Explain more clearly why certain results are novel or unexpected, and thus interesting to other researchers. Summarize a few "best practices" for researchers in the field looking to pretrain their model.
- Include more insights and explanations for why some methods perform better than others.


Please see additional questions below.

- "Furthermore, we consider scaling the PDE solution (Scale), an approach similar to a color distortion, in which the PDE solution values are multiplied by a random constant."

  What does "similar to a color distortion" mean?

- It is unclear which parts of the Methods are novel and which parts are not. E.g., which augmentations in Section 3.1 have been used for PDE before? Which pretraining strategies in Section 3.2 have been used for PDE before?

- "This is implemented by patching the solution in space and time, randomly replacing masked patches with a learnable mask token, and regressing the true solution; we refer to this strategy as Masked."

  It appears this pretraining strategy only applies to transformer models because of the use of mask tokens. Could the authors explain how a "mask token" could be used in a CNN? Also, is this proposed by Zhou & Farimani (2024b)?

- "Both works require the use of a specific encoder along with the neural operator backbone; to adapt these strategies to our experimental setup we consider directly pretraining the neural operator contrastively with these strategies."

  Could the authors explain in more detail what the "specific encoder" is? Why is it used in previous works? And how did the authors mitigate the need for this encoder?

- Tables 2 and 3:
The authors only showed what the best-performing method is and its performance for each case. It would be more informative to show the performance of all methods, so that the gap between methods is clear.
Also, is there any pattern in when each method performs best? Is there any rule of thumbs extracted from these results that can inform model training?

**Strengths And Weaknesses:**

**Strengths**

The paper performed extensive experiments, covering a wide range of models, PDE problems, pretraining and augmentation methods, and downstream problems.

**Weaknesses**

In my humble opinion, many conclusions are primarily empirical and could benefit from more analysis. Some conclusions might be anticipated and could benefit from better justification of interest. Some claims are not fully supported. Please see examples below.

1. The main conclusion that "directly adapting computer vision methods to the physics domain generally results in poor performance" might be anticipated given the different stuctures of the problems. It is unclear whether a more thoughtful adaptation considering the structure of PDE problems would work better. Although negative results are important, the reviewer is not sure whether it is already known to the field that CV methods would not work well in PDE. Also, the motivation for adapting CV methods could be better justified. Why focus on CV, rather than methods in other ares, e.g. natural language processing?

2. The paper could benefit from more insight and theoretical explanations into why some methods work better than others. E.g., why does shift augmentation outperform others?

3. The authors found that transfer learning + shift augmentation performs best in general. However, it is unclear how combination of multiple augmentations will perform, which is most common in machine learning practice.

4. Table 4: The conclusion that "Pretraining is more beneficial when downstream data is similar to pretraining data." is not fully supported. Two out of four methods (OFormer and Unet) showed contraditory results.

5. The conclusions that "Pretraining is more beneficial when downstream data is scarce" and "Pretraining is more beneficial when downstream data is similar to pretraining data" in Section 5.4 seem to be well-known in machine learning.

6. Some claims are not supported by data. E.g.,

   "Empirically, SpaceSort does not perform well, so we omit this strategy from our results."

   "However, these methods did not seem to show significant improvements over no pretraining, as such, the results are omitted from the paper."

   Could the authors provide more evidence for these claims, either by referring to previous work or by providing their own results?

---

> ### Author Response · Authors · 2024-07-25
> **Response to Reviewer yf78**
>
> Thank you for your thoughtful review of our paper. Please find our comments and revisions below to address your concerns about our work. We hope that this clears up any questions or concerns you have, but if there is anything else please do not hesitate to respond or reach out.
>
> Weaknesses:
>
> 1. We agree that CV and physics problems have different structures, we believe it is not immediately obvious that CV methods would not work well with PDEs. For example, denoising is used in both CV and PDEs, and has been shown to improve performance in both fields. We believe this motivates exploring additional CV-inspired pretraining strategies. We look at CV methods because images and video are directly analogous to PDE data (which is discretized on a grid). Additionally, we consider masked pretraining strategy, which is similar to the NLP pretraining strategy of masked token prediction. To the best of our knowledge, these methods have not been directly compared before in this context. We also consider more thoughtful adaptations such as using Lie data augmentations that are designed to respect the underlying physics and could be similar to CV data augmentation techniques, as well as Physics Informed Contrastive Learning (PICL), a pretraining strategy that uses domain knowledge of PDEs with Generalized Contrastive Loss, that was originally developed for video applications.
>
> 2. Thank you for the suggestion, we agree that some points could use some more explanations. We add theoretical explanations in a new discussion section (Section 6). As for the shift augmentation, it is the only augmentation that conserves the dynamics across our data sets, since shifted solutions are still valid solutions to the governing PDE as physical problems are spatially invariant. We believe the preserved dynamics leads to improved performance in downstream tasks.
>
> 3. We agree that the effect of a combination of augmentations during pretraining is still unknown, however based on prior works [1], the trend is that a combination of augmentations could help. However, testing all these combinations would be computationally infeasible, and the optimal combination could vary based on the PDE or model. As as result, we leave this optimization to practitioners looking to leverage pretraining for their specific problems, which is a common practice for those looking to develop their own machine learning models and instead make the argument that incorporating data augmentations can generally help. We have added a discussion on combining multiple data augmentations and the possibility of it improving performance in section 6.2. [1] Johannes Brandstetter, Max Welling, Daniel E. Worrall, Lie Point Symmetry Data Augmentation for Neural PDE Solvers, https://arxiv.org/abs/2202.07643
>
> 4. We agree and thank you for noticing this. We have updated our claim from “Pretraining is more beneficial when downstream data is similar to pretraining data.” to “Pretraining improves performance for both in and out-of-distribution downstream tasks.”
>
> 5. We agree that it is well known that pretraining tends to help when downstream data is scarce. We noted this in section 5.3: “In general, we observe a trend in which the improvement of pretrained models diminishes as the number of fine-tuning samples increases, which is expected as fine-tuning data approaches the pretraining dataset size.” For generalization in section 5.4, we note an improvement in relative performance when generalizing to out-of-distribution data that is likely due to sharing the underlying dynamics present in PDE systems. More broadly, although these claims are well-known in general, we empirically validate them, as well as provide useful context, nuances, and theoretical explanations for interpreting these claims within the PDE domain.
>
> 6. Thank you for bringing this concern to our attention, we have included our preliminary results to support these claims in Appendix E.

---

> > ### Author Response · Authors · 2024-07-25
> > **Response to Reviewer yf78**
> >
> > Requested Changes
> >
> > 1. We have added “We focus on vision-inspired strategies because the point-wise representation of system values on a grid for a PDE is analogous to pixel-based representations of images, and the temporal structure of PDE system evolution is analogous to videos.” to the introduction.
> >
> > 2. Thank you for the suggestion. We have added a discussion section with subsection 6.2 Best Practices. That makes suggestions on which strategies to use, as well as an explanation of why.
> >
> > 3. We have provided insight on why some methods perform better than others in our discussion section, Section 6.
> >
> > Questions
> >
> > 1. We are referring to a common data augmentation in the CV domain that is referred to as color jitter or color distortion whereby the hue, saturation, contrast, or lightness of an image is randomly changed. Although these visual characteristics do not have direct counterparts in the physics domain, changing these characteristics of a color could be interpreted as scaling how colorful, vivid, contrasting, or bright the color appears. Scaling the magnitude of a PDE solution could also be interpreted in a similar manner. We clarify this point in Section 3.1.2.
> >
> > 2. All of the presented data augmentations have been applied to PDEs and we cite the corresponding papers, however, this paper is the first time that they have been compared against each other on different PDEs and models on different pretraining tasks. As for pretraining strategies, besides transfer learning (which is the prevailing pretraining method in prior works), and PICL, all the presented methods have not been applied to PDEs in how they are presented in our work. More broadly, even if these methods have been individually investigated before, this is the first work to compare them against each other and draw conclusions about pretraining as a whole for physical problems.
> >
> > 3. In our work, we use a more general definition of a mask token, specifically, we define a learnable parameter in physical space rather than the embedding space. This allows patches of physical data to be replaced by a learnable mask token, which a neural operator can then process. This is in contrast to transformer methods that replace embedded tokens with a mask token. We recognize that this could be confusing so we appreciate the comment and we have made a clarification in Section 3.2.2.
> >
> > 4. Previous contrastive works generally focus on pretraining an encoder to extract contextual information from physical problems through clustering samples in a latent space. The encoder is then used to inform a neural operator by conditioning on an encoded representation of the physical data. A drawback of this approach is that a conditioning mechanism needs to be tailored for each neural operator, for example, a transformer architecture may have a different conditioning mechanism than a CNN architecture. To mitigate this, we investigated directly using the neural operator with the contrastive loss function, then during fine-tuning using the physics prediction loss. We add a clarification in Section 3.2.3.
> >
> > 5. We agree that the overall picture of pretraining for PDEs is more nuanced than showing the best pretraining method, and we present the full results comparing all methods in the Appendix. In the main body, we extracted some broad claims about these granular results, and added a section (Section 6) to discuss patterns and best practices.

---

> > > ### Comment · Reviewer_yf78 · 2024-08-04
> > >
> > > Thank the authors for addressing all my questions.

---

### Review · Reviewer_rDWh · 2024-07-09

**Summary Of Contributions:**

This paper evaluates a range of pretraining and data augmentation
methods for neural operators, testing a variety of network
architectures against a range of target physics. The results may
provide some insight into which pretraining approaches work well for
PDE tasks, which could help guide future work. The authors include
some experiments testing the impact of larger vs smaller fine-tuning
datasets and in- vs. out-of-distribution samples in the fine-tuning
phase vs. pretraining.

**Audience:**

Yes

**Broader Impact Concerns:**

No concerns that I can identify

**Claims And Evidence:**

Yes

**Requested Changes:**

No particularly critical changes in my opinion, mostly some additions
I believe would add clarity.

1. It would be helpful to have more detail on network architectures in
   appendix D.2. As a particular example, for the U-net there are a
   variety of possible structures, and a more complete description
   would be useful.
2. Similarly I believe it would also be helpful to provide some
   details of the coefficient distributions used to generate the
   datasets in appendix D.1.
3. A minor question: in appendix D.3 you note that networks are
   "generally trained for 200 epochs..." were there exceptions? If
   not, it would help to make this a bit more definitive.
4. Will the code used for the experiments be available? I don't
   believe it is critical for the review, but would be helpful for
   readers who want full details of the experiments beyond what can be
   communicated in the paper.
5. For the augmentation experiments in Table 3 and appendix B.2 you
   mention the results are compared against a model without
   pretraining? In that case, does it reflect the effect of the data
   augmentation independent of the pretraining step? For example, for
   Burgers-Unet Table 3b lists a ~10% improvement but based on the
   table in the appendix, the improvement over plain Transfer appears
   to be more modest. It does appear that the augmentation provides
   some further benefit, but this could use some additional
   discussion.
6. Another very minor note: in appendix D.4 you note that the scale
   augmentation does not respect physics for Burgers. You mention this
   as a possible issue in section 3.1.2. It may be good to briefly
   note that this does in fact affect one of the systems you consider.
7. Another question on the construction of the fine-tuning datasets:
   Looking at appendix D.5 it is a bit clear whether you resampled the
   smaller fine-tuning sets independently for each independent network
   training run, or whether the sets were fixed for each size? Were
   the larger sets chosen as strict supersets of smaller data sets? I
   think this could do with a bit more clarity since depending on the
   structure it could add more variance to some of the results.

**Strengths And Weaknesses:**

**Strengths**
- The paper is generally clearly written and includes a good introduction to the problem and discussion of related works
- Good variety of experimental configurations (target physics, augmentations, pretraining strategies, etc.)
- The results are well categorized to help the reader understand them despite the varied experimental configuration

**Weaknesses**
- The paper might benefit from additional detail on model architectures
- Some details of the experiments could also be more explicitly described
- A few tests might have benefited from more independently seeded
  networks in cases with high variance (but understandably there are limits to what is feasible)

---

> ### Author Response · Authors · 2024-07-25
> **Response to Reviewer rDWh**
>
> Thank you for your interest in our work, and we appreciate your careful reading of our manuscript to identify points of clarification. We have revised our paper to address them and commented on these below, but if there is still anything unclear please do not hesitate to reach out.
>
> Weaknesses
>
> 1. Thank you for the comment, we have added some additional details in Appendix D.2. Furthermore, we plan on releasing code when the paper is published so that the implementations and details can be seen.
>
> 2. We have added additional details about hyperparameters used during training in Appendix D.4 and D.5. Additionally, we plan on releasing the code used to reproduce the experiments so that all the configs and experimental details can be seen.
>
> 3. We recognize that many of the experiments have high variance, but will note that there are likely other factors affecting it other than the seed. In particular, the fine-tuning dataset is randomly sampled from a larger dataset; for example, when fine-tuning with 100 samples, we sample 100 samples from a larger unseen set of 1024 samples, which will vary across experiments. As such, although the variance is high when combining seeds, the trends observed within a seed (i.e. within a sampled dataset), follow trends consistent with the mean of the overall results.
>
> Requested Changes:
>
> 1. Thank you for the comment, we have added additional details on model architectures in Appendix D.2.
>
> 2. We added details on the coefficient distributions in sections 4.1.1 and 4.1.2, hopefully those would be sufficient.
>
> 3. At 200 epochs all pretraining losses converge, however, for Binary pretraining, which is a much simpler task, the loss converges much earlier. To save on computation we pretrain in this setting to fewer epochs. We have added a note about this in the Appendix.
>
> 4. We plan on releasing code, and have provided an anonymized version in the supplementary zip file if you’d like to view it.
>
> 5. You are correct about this, indeed when writing the paper, we had to balance conciseness with the detailed results in the Appendix. The percentages presented in the main body in Table 3 based on data augmentations are compared to no pretraining, which would make it dependent on the pretraining strategy. Considering the effect of the pretraining strategy would make the summary statistics more complicated as there are different improvements based on different pretraining strategies, so we elected to leave that analysis to the appendix. However, we will add a discussion about this in Section 5.2 in the main body as it is an important note.
>
> 6. Thank you for the suggestion, we have added a note about this for Burgers equation in section 3.1.2.
>
> 7. We agree that this was a little unclear, and will clarify that we independently re-sampled each fine-tuning set for each training run in Appendix D.5. The larger sets were not strict supersets of the smaller data sets, but they are also independently re-sampled from the same pool for each training run. We have added a note about this in the Appendix.

---

> > ### Comment · Reviewer_rDWh · 2024-08-26
> >
> > Thank you for your revisions and your answers to my questions. I appreciate the clarifications and I believe the additional details have helped enhance the clarity of the paper.

---

### Review · Reviewer_qE5d · 2024-07-19

**Summary Of Contributions:**

The paper functions both as a survey of existing methods of transfer learning for neural PDE autoregressive operator (i.e. forward propagator) learning, and as novel research comparing SOTA methods under an interesting treatment condition.

**Audience:**

Yes

**Claims And Evidence:**

No

**Requested Changes:**

## Critical changes

1. At least mention the wider field of transfer learning and provide a rationale as to why generic transfer learning and data augmentation theory results are not helpful to this particular sub-field.
2. Do something to address the lack of reproducibility standardisation in the data set. In order of preference I would suggest

   1. Use the existing standard pde datasets ( [pdearena](https://github.com/pdearena/pdearena)/ [PDEBench](https://github.com/pdebench/PDEBench/)) or if they are not suitable,
   2. submit pull requests to the maintainers so that they are suitable, or if this is not productive,
   3. publish the code or datasets used in the results so that future works are comparable with this one

## Recommended changes

1. if you could apply any of the existing transfer learning theory to explain the results here that would be valuable.
1. Use some of the theoretical results from the wider field, about optimal transfer methods etc (adversarial learning! invariant learning)

There are several taxonomies and communities of transfer learning who might be useful for this:

  * [thuml/A-Roadmap-for-Transfer-Learning](https://github.com/thuml/A-Roadmap-for-Transfer-Learning#domain-adaptation)
  * [zhaoxin94/awesome-domain-adaptation](https://github.com/zhaoxin94/awesome-domain-adaptation)
  * [http://transferlearning.xyz](http://transferlearning.xyz/)

A partial listing of recent and highly-cited works in transfer learning is

```
@article{ImanReview2023,
  title = {A {{Review}} of {{Deep Transfer Learning}} and {{Recent Advancements}}},
  author = {Iman, Mohammadreza and Rasheed, Khaled and Arabnia, Hamid R.},
  year = {2023},
  month = mar,
  journal = {Technologies},
  volume = {11},
  number = {2},
  eprint = {2201.09679},
  pages = {40},
  doi = {10.3390/technologies11020040},
  archiveprefix = {arXiv}
}

@misc{JiangTransferability2022,
  title = {Transferability in {{Deep Learning}}: {{A Survey}}},
  shorttitle = {Transferability in {{Deep Learning}}},
  author = {Jiang, Junguang and Shu, Yang and Wang, Jianmin and Long, Mingsheng},
  year = {2022},
  month = jan,
  number = {arXiv:2201.05867},
  eprint = {2201.05867},
  publisher = {arXiv},
  archiveprefix = {arXiv}
}

@misc{KaddourCausal2022,
  title = {Causal {{Machine Learning}}: {{A Survey}} and {{Open Problems}}},
  shorttitle = {Causal {{Machine Learning}}},
  author = {Kaddour, Jean and Lynch, Aengus and Liu, Qi and Kusner, Matt J. and Silva, Ricardo},
  year = {2022},
  month = jul,
  number = {arXiv:2206.15475},
  eprint = {2206.15475},
  publisher = {arXiv},
  doi = {10.48550/arXiv.2206.15475},
  archiveprefix = {arXiv}
}

@article{LiuDeep2022,
  title = {Deep {{Unsupervised Domain Adaptation}}: {{A Review}} of {{Recent Advances}} and {{Perspectives}}},
  shorttitle = {Deep {{Unsupervised Domain Adaptation}}},
  author = {Liu, Xiaofeng and Yoo, Chaehwa and Xing, Fangxu and Oh, Hyejin and Fakhri, Georges El and Kang, Je-Won and Woo, Jonghye},
  year = {2022},
  month = aug,
  journal = {APSIPA Transactions on Signal and Information Processing},
  volume = {11},
  number = {1},
  publisher = {Now Publishers, Inc.},
  doi = {10.1561/116.00000192}
}

@misc{LiuOutOfDistribution2023,
  title = {Towards {{Out-Of-Distribution Generalization}}: {{A Survey}}},
  shorttitle = {Towards {{Out-Of-Distribution Generalization}}},
  author = {Liu, Jiashuo and Shen, Zheyan and He, Yue and Zhang, Xingxuan and Xu, Renzhe and Yu, Han and Cui, Peng},
  year = {2023},
  month = jul,
  number = {arXiv:2108.13624},
  eprint = {2108.13624},
  publisher = {arXiv},
  doi = {10.48550/arXiv.2108.13624},
  archiveprefix = {arXiv}
}

@inproceedings{MansourDomain2009,
  title = {Domain {{Adaptation}}: {{Learning Bounds}} and {{Algorithms}}},
  shorttitle = {Domain {{Adaptation}}},
  booktitle = {{{COLT}}},
  author = {Mansour, Yishay and Mohri, Mehryar and Rostamizadeh, Afshin},
  year = {2009},
  month = feb,
  eprint = {0902.3430},
  publisher = {arXiv},
  doi = {10.48550/arXiv.0902.3430},
  archiveprefix = {arXiv}
}

@article{NiuDecade2020,
  title = {A {{Decade Survey}} of {{Transfer Learning}} (2010--2020)},
  author = {Niu, Shuteng and Liu, Yongxin and Wang, Jian and Song, Houbing},
  year = {2020},
  month = oct,
  journal = {IEEE Transactions on Artificial Intelligence},
  volume = {1},
  number = {2},
  pages = {151--166},
  doi = {10.1109/TAI.2021.3054609}
}

@misc{TanSurvey2018,
  title = {A {{Survey}} on {{Deep Transfer Learning}}},
  author = {Tan, Chuanqi and Sun, Fuchun and Kong, Tao and Zhang, Wenchang and Yang, Chao and Liu, Chunfang},
  year = {2018},
  month = aug,
  number = {arXiv:1808.01974},
  eprint = {1808.01974},
  publisher = {arXiv},
  doi = {10.48550/arXiv.1808.01974},
  archiveprefix = {arXiv}
}

@article{WangGeneralizing2023,
  title = {Generalizing to {{Unseen Domains}}: {{A Survey}} on {{Domain Generalization}}},
  shorttitle = {Generalizing to {{Unseen Domains}}},
  author = {Wang, Jindong and Lan, Cuiling and Liu, Chang and Ouyang, Yidong and Qin, Tao and Lu, Wang and Chen, Yiqiang and Zeng, Wenjun and Yu, Philip S.},
  year = {2023},
  month = aug,
  journal = {IEEE Transactions on Knowledge and Data Engineering},
  volume = {35},
  number = {8},
  pages = {8052--8072},
  doi = {10.1109/TKDE.2022.3178128}
}

@book{WangIntroduction2023,
  title = {Introduction to {{Transfer Learning}}: {{Algorithms}} and {{Practice}}},
  shorttitle = {Introduction to {{Transfer Learning}}},
  author = {Wang, Jindong and Chen, Yiqiang},
  year = {2023},
  series = {Machine {{Learning}}: {{Foundations}}, {{Methodologies}}, and {{Applications}}},
  publisher = {Springer Nature},
  address = {Singapore},
  doi = {10.1007/978-981-19-7584-4},
  isbn = {978-981-19758-3-7 978-981-19758-4-4}
}

@misc{WilsonSurvey2020,
  title = {A {{Survey}} of {{Unsupervised Deep Domain Adaptation}}},
  author = {Wilson, Garrett and Cook, Diane J.},
  year = {2020},
  month = feb,
  number = {arXiv:1812.02849},
  eprint = {1812.02849},
  publisher = {arXiv},
  doi = {10.48550/arXiv.1812.02849},
  archiveprefix = {arXiv}
}

@book{YangTransfer2020,
  title = {Transfer {{Learning}}},
  author = {Yang, Qiang and Zhang, Yu and Dai, Wenyuan and Pan, Sinno Jialin},
  year = {2020},
  edition = {1},
  publisher = {Cambridge University Press},
  address = {Cambridge},
  doi = {10.1017/9781139061773},
  isbn = {978-1-107-01690-3}
}

@inproceedings{ZhangBridging2019,
  title = {Bridging {{Theory}} and {{Algorithm}} for {{Domain Adaptation}}},
  booktitle = {Proceedings of the 36th {{International Conference}} on {{Machine Learning}}},
  author = {Zhang, Yuchen and Liu, Tianle and Long, Mingsheng and Jordan, Michael},
  year = {2019},
  month = may,
  pages = {7404--7413},
  publisher = {PMLR}
}

@article{ZhangTransfer2024,
  title = {Transfer {{Adaptation Learning}}: {{A Decade Survey}}},
  shorttitle = {Transfer {{Adaptation Learning}}},
  author = {Zhang, Lei and Gao, Xinbo},
  year = {2024},
  month = jan,
  journal = {IEEE Transactions on Neural Networks and Learning Systems},
  volume = {35},
  number = {1},
  pages = {23--44},
  doi = {10.1109/TNNLS.2022.3183326}
}

@misc{ZhuangComprehensive2020,
  title = {A {{Comprehensive Survey}} on {{Transfer Learning}}},
  author = {Zhuang, Fuzhen and Qi, Zhiyuan and Duan, Keyu and Xi, Dongbo and Zhu, Yongchun and Zhu, Hengshu and Xiong, Hui and He, Qing},
  year = {2020},
  month = jun,
  number = {arXiv:1911.02685},
  eprint = {1911.02685},
  publisher = {arXiv},
  doi = {10.48550/arXiv.1911.02685},
  archiveprefix = {arXiv}
}
```

**Strengths And Weaknesses:**

## Strengths

Well-explained methods, including comparison to generic image regression tasks, good illustrative diagrams.

* Useful literature review, including many references in this domain of which I at least was not aware
* interesting auxiliary question about the interactions of data augmentation and transfer learningn

## Weaknesses

* The empirical results are not supported by much theory
* This paper is at the intersection of two fields, transfer learning and neural operator learning, and it faces a common challenge for such multi-field papers: how to satisfy readers from both fields. I am sympathetic to the fact that this is difficult to do, but I nonetheless think the authors need to do make some changes to be effective here.

* Unclear reproducibility and poor positioning in relation to existing literature on benchmarking PDE tasks;
  The paper does not, it seems, use directly, or extend the existing PDE benchmark datasets

  * [pdearena](https://github.com/pdearena/pdearena)
  * [PDEBench](https://github.com/pdebench/PDEBench/)

  Maybe these methods are not useful in their case, but the "shape" of the data used in the pre-training, covariate-shift and fine tuning does look a lot like the stuff in the standard data sets, which leads to the question of why not use the exist methods rather than introduce new data sets

* The paper does not put its results in the context of pretraining, domain adaptation, transfer-learning  fine-tuning generally, treating the case of neural operator learning as essentially an isolated field. It might be true that fine-tuning and transfer learning have little to add to the special case of neural operators, but this looks like a non-trivial assertion. The main section addressing this relationship seems to be sections 2.1 and 2.4:

  >Many past works consider transferring knowledge between PDE parameters and domains as a form of pre-
  >training. These works often design specific architectures that are tailored for transferring weights or layers
  >between tasks. For example, Goswami et al. (2022) design task-specific layers of a DeepONet to be used
  >with different domains of 2D Darcy Flow and Elasticity problems. Another approach proposed by Tripura &
  >Chakraborty (2023) is to design different operators that learn specific PDE dynamics and combine these in
  >a mixture of experts approach, motivated by the observation that PDEs can often be compositions of each
  >other. To address the issue of transferring between physical domains that can have different numbers of vari-
  >ables, Rahman et al. (2024) extend positional encodings and self-attention to different codomains/channels

  >Additional past work considers adapting meta-learning (Finn et al., 2017) paradigms from the broader deep
  >learning community to the PDE domain. Zhang et al. (2023a) consider a direct adaptation of model-agnostic
  >meta-learning to PDE tasks, while Yin et al. (2021) and Kirchmeyer et al. (2022) apply novel losses and
  >architectures to maximize shared learning across different tasks. Following in-context learning trends of
  >transformer models (Dong et al., 2023), Chen et al. (2024) and Yang et al. (2023) explore using in-context
  >learning to prompt models with PDE solutions to generalize to unseen PDE coefficients

  We've already admitted that the wider field of transfer learning might have something to say about operator learning in particular, so now we must wonder why their results are not used to inform the interesting questions of the paper with regards to data augmention.

  The taxonomy in Table 1 suggests that the community of practitioners who look at both transfer learning and PDE operator learning might be somewhat separate, so this separation probably reflects the conventions of the field.

---

> ### Author Response · Authors · 2024-07-25
> **Response to Reviewer qE5d**
>
> Thank you for your thoughtful review of our paper and your effort in compiling resources to help us. We appreciate the interesting viewpoints from the broader transfer learning field, and is something that we had not considered before. Please find a response to your comments below and the corresponding revisions to our paper.
>
> Weaknesses
>
> 1. Thank you for the suggestion. We have discussed some hypotheses and underlying theory as to why we believe the results are observed in Section 6.1, however we believe a full theoretical treatment of the problem is beyond the scope of this work.
>
> 2. We appreciate the comment and the help in providing the citations. We have added a section (Section 2.5) that incorporates works from the transfer learning domain that are relevant to PDE pretraining as well as discusses how transfer learning works could fit into the PDE domain.
>
> 3. We agree that the generated data is similar to common benchmarks in that they both model phenomena on a regular grid with periodic boundary conditions, however, the main difference with our dataset is that PDE coefficients are sampled uniformly, rather than being set to a discrete value. For example, PDEBench provides Advection data with 8 arbitrarily determined wave speeds, but we opt to uniformly sample this coefficient from an interval which encompasses these predetermined values. This results in a more challenging task, as different data samples will display different dynamics. We believe that this is more representative of real-world uses of transfer learning, since the downstream application will likely not have discrete PDE coefficients that one can determine a priori, and in general PDE coefficients will vary continuously (e.g. fluids have different viscosities, materials have different thermal diffusivities, etc.).
>
>     Another reason to generate our own data is to be able to control the covariate shift between a pretraining and fine-tuning dataset. Current PDE benchmarks have a wide range of data, however, they were not made for the purpose of studying transfer performance when changing PDE parameters. As a result, much of the benchmark data is extremely different from each other, which reflects the original intention for the benchmark to quantify a model’s performance across a wide range of PDE tasks.
>
>     In relation to existing literature, the field of neural operators is quite new and as a result, early works benchmark on the easier case of predicting phenomena under constant or discretely determined coefficients. However, as the performance of neural operators increases, newer works have begun to address this practical limitation by using more diverse and challenging datasets that have a larger variation of coefficients, and thus can be applied to a wider variety of unseen PDE tasks [1, 2, 3, 4]. We anticipate that this trend may become more popular in future works, so we opt to use this setup in our work. With regards to reproducibility, we plan on providing the code to generate the necessary data when the paper is published. The physical phenomena modeled by the data are relatively simple and well-studied and as a result, the code can be run without any special hardware and the findings from the paper can be loosely extrapolated to many similar datasets in other benchmarks.
>
> - 1. Shashank Subramanian, Peter Harrington, Kurt Keutzer, Wahid Bhimji, Dmitriy Morozov, Michael Mahoney, Amir Gholami. Towards Foundation Models for Scientific Machine Learning: Characterizing Scaling and Transfer Behavior. https://arxiv.org/abs/2306.00258
> - 2. Hang Zhou, Yuezhou Ma, Haixu Wu, Haowen Wang, Mingsheng Long. Unisolver: PDE-Conditional Transformers Are Universal PDE Solvers. https://arxiv.org/abs/2405.17527
> - 3. Maximilian Herde, Bogdan Raonić, Tobias Rohner, Roger Käppeli, Roberto Molinaro, Emmanuel de Bézenac, Siddhartha Mishra. Poseidon: Efficient Foundation Models for PDEs. https://arxiv.org/abs/2405.19101
> - 4. Benedikt Alkin, Andreas Fürst, Simon Schmid, Lukas Gruber, Markus Holzleitner, Johannes Brandstetter. Universal Physics Transformers: A Framework For Efficiently Scaling Neural Operators. https://arxiv.org/abs/2402.12365
>
> 4. We believe you are right that PDE operator learning is generally a more isolated field in deep learning and that it is nevertheless important to consider how the broader transfer learning field can inform this paper. We have added content discussing findings from the broader transfer learning field and how they are applied to the results observed from PDE operator learning in Section 6.1.

---

> > ### Author Response · Authors · 2024-07-25
> > **Response to Reviewer qE5d**
> >
> > Requested Changes:
> >
> > Critical changes
> >
> > 1. We believe that the wider field of transfer learning could have some insights to inform PDE learning, and we’ve added the corresponding discussion in Sections 2.5 and 6.
> >
> > 2. We appreciate your thought in providing a list of suggestions and we agree that reproducibility is important. We will publish the code used to generate data for the paper, however, for reasons outlined in #3 above, using previous benchmarks would be challenging for this work. Additionally, proposing changes to benchmarks would also be challenging due to needing to change the way PDE parameters are sampled in PDEBench or PDEArena, which fundamentally affects the data generation process.
> >
> > Recommended changes
> >
> > 1. We appreciate the suggestion, we have added some discussion about this in Section 6.1.
> >
> > 2. Thank you for the suggestion, however this could be out of scope for the current work. We envisioned the paper more as a survey, comparison, and best practices of pretraining methods for PDE modeling, rather than proposing completely new transfer methods, however we have added a discussion about optimal/novel transfer methods as a promising future direction in both Sections 2.5 and 6.1.

---

> > > ### Comment · Reviewer_qE5d · 2024-09-06
> > >
> > > I think the authors for their responsiveness. The authors have taken a step in the direction I suggested, which I appreciate. I still don't find the connection to the wider literature strong, or profound, but it does what it says on the tin.

---

### Review · Reviewer_PjtC · 2024-07-24

**Summary Of Contributions:**

This is a well-written experimental paper. Although there are many highlights such as comparing various pretraining methods without optimizing architecture choices to characterize pretraining dynamics on different models and datasets, pretraining performance can be further improved by using data augmentations, and pretraining is additionally beneficial when fine-tuning in scarce data regimes or when generalizing to downstream data similar to the pretraining distribution.

**Audience:**

No

**Broader Impact Concerns:**

I do not think that this work has important influences.

**Claims And Evidence:**

No

**Requested Changes:**

The authors should rewrite the manuscript to make the key point clearer.

**Strengths And Weaknesses:**

However, there are some essentially weak points. The content is too distracting so that the person who read it is hard to capture the key points. Although the authors do so many experiments, I still cannot get the point. If the author can re-written the content and show the experiment clearly, i.e. where is strong by adding pretraining.

---

> ### Author Response · Authors · 2024-07-25
> **Response to Reviewer PjtC**
>
> Thank you for the review of our paper. We have provided some comments below and have revised our paper accordingly.
>
> Weaknesses and Requested Changes
>
> We appreciate your comments regarding the clarity of the paper. To address this, we have added a new discussion section (Section 6) aimed at summarizing the results and providing the key points of the paper.
> More broadly, the paper has many results by nature of comparing different pretraining strategies, models, and datasets. We have tried our best to summarize these to the main conclusions of:
> - Pretraining improves model performance, however, the optimal strategy is dependent on the model architecture and dataset.
> - Data augmentations consistently help when pretraining models.
> - Pretraining is beneficial when there is a limited number of fine-tuning samples
> - Pretraining has some ability to generalize to unseen domains and problems.
>
> These claims are based on empirical results and some theoretical justifications, and if this is still unclear, please let us know

---

### Decision · Action_Editor_8KNY · 2024-09-22

**Recommendation:** Accept as is

**Comment:**

The reviewers appreciated the effort put in this empirical investigation and praised it as it can be very interesting for practitioners looking fo squeezing extra performance out of neural operators. At the same time, they have raised a number of concerns, the major ones being the lack of theory, unclear reproducibility, and missing comparisons to existing literature on benchmarking neural PDE predictors.

During the rebuttal, the authors addressed well the last two points and managed to flip some of the reviewers, now providing these recommendations: accept, leaning accept and leaning reject. The criticism still remaining is that this work is essentially empirical and has no supporting theory. I agree this is the case, but this cannot be a reason to reject the paper, as its claims and contributions are supported by enough (empirical!) evidence.

The manuscript could be accepted as is, but I require the authors to make another pass spotting typos and fixing the width of Table 1.

**Audience:**

The main contribution is definitely helpful for people working in physics-inspired ML, and therefore of clear interest for the TMLR audience.

**Claims And Evidence:**

The authors provide an empirical investigation on how to (pre-)train neural operators to solve PDEs. To this end, they consider a range of pre-training and data augmentation techniques and furthermore investigate the impact of sample size when fine-tuning for in- and out-of-distribution scenarios. The outcome is not a clear-cut message, but the finding that the effect of these strageies is highly dependent on model and dataset chosen. On the positive side, transfer learning and data augmentation seem to be beneficial in general.

---

> ### Comment · Action_Editor_8KNY · 2024-09-28
> **Please check the format of references in Table 1**
>
> Dear authors,
>
> I appreciated that you tried to fix the width of Table 1, but using numbers for citations (only there) might break the TMLR format (and if someone prints the article on paper, it is impossible to recover the citation).
> I suggest you revert back to the previous scenario and save space by e.g., rotating by 90 degrees the labels in the first column, and trim a bit the third column.

---

> > ### Author Response · Authors · 2024-09-28
> > **Fixed Table 1 Formatting**
> >
> > Thank you for the suggestion, the Table has been fixed!